# Projection-specific integration of convergent thalamic and retrosplenial signals in the presubicular head direction cortex

Louis Richevaux[1]*, Dongkyun Lim[1,2], Mérie Nassar[1], Léa Dias Rodrigues[1], Constanze Mauthe[1], Ivan Cohen[3], Nathalie Sol-Foulon[4], Desdemona Fricker[1,2]*

[1]Université Paris Cité, CNRS, Integrative Neuroscience and Cognition Center, Paris, France; [2]Université Paris Cité, CNRS, Saints-Pères Paris Institute of the Neurosciences, Paris, France; [3]Sorbonne Université, INSERM, CNRS, Neuroscience ParisSeine, Institut de Biologie Paris Seine, Paris, France; [4]Sorbonne Université, INSERM, CNRS, Paris Brain Institute, ICM, Pitié-Salpêtrière Hospital, Paris, France

**\*For correspondence:**
louis.richevaux@parisdescartes.fr (LR);
desdemona.fricker@parisdescartes.fr (DF)

**Competing interest:** The authors declare that no competing interests exist.

## eLife Assessment

This **valuable** study combines anatomical tracing and slice physiology to examine how anterior thalamic and retrosplenial inputs converge in the presubiculum, a key region in the navigation circuit. The authors show that near-simultaneous co-activation of retrosplenial and thalamic inputs drives supra-linear presubiculum responses, revealing a potential cellular mechanism for anchoring the brain's head direction system to external visual landmarks. Their thorough experimental approach and analyses provide **convincing** evidence for the cellular basis of how the brain's internal compass may be anchored to the external world, laying the groundwork for future experimental testing in vivo.

**Abstract** Head direction (HD) signals function as the brain's internal compass. They are organized as an attractor and anchor to the environment via visual landmarks. Here, we examine how thalamic HD signals and visual information from the retrosplenial cortex combine in the presubiculum. We find that monosynaptic excitatory connections from anterior thalamic nucleus and from retrosplenial cortex converge on single layer 3 pyramidal neurons in the dorsal portion of mouse presubiculum. Independent dual-wavelength photostimulation of these inputs in slices leads to action potential generation preferentially for near-coincident inputs, indicating that layer 3 neurons can transmit a visually matched HD signal to medial entorhinal cortex. Layer 4 neurons, which innervate the lateral mammillary nucleus, form a second step in the association of HD and landmark signals. They receive little direct input from thalamic and retrosplenial axons. We show that layer 4 cells are excited di-synaptically, transforming regular spiking activity into bursts of action potentials, and that their firing is enhanced by cholinergic agonists. Thus, a coherent sense of orientation involves projection-specific translaminar processing in the presubiculum, where neuromodulation facilitates landmark updating of HD signals in the lateral mammillary nucleus.

## Introduction

The head direction (HD) system functions as the brain's compass system. It is distributed across several interconnected brain structures and hierarchically organized from the brainstem to the lateral mammillary nucleus (LMN; *Stackman and Taube, 1998*), the anterior thalamic nuclei (ATN; *Taube, 1995*), and the dorsal presubiculum (PrS; *Ranck, 1984*; *Taube et al., 1990a*). Modeling work suggests the HD system functions as a ring attractor, where the population of HD neurons is arranged in a one-dimensional manifold (*Blair and Sharp, 1995*; *McNaughton et al., 2006*; *Skaggs et al., 1995*). The activity of thalamic and PrS HD neurons is correlated across different brain states, even during sleep, independently of sensory cues (*Peyrache et al., 2015*). The bump-like activity dynamics of an attractor network can maintain HD signals based on excitatory-excitatory or excitatory-inhibitory interactions (*Knierim and Zhang, 2012*; *Simonnet et al., 2017*). However, HD signals may drift in the dark (*Taube et al., 1990b*; *Zugaro et al., 2003*). For precise navigation, internally generated information on head direction must be combined with awareness of location in an environment. Mechanisms underlying the combination of egocentric head direction signals with anchoring to allocentric landmarks remain to be clarified.

Head direction and visual landmark signals may be integrated in the presubiculum (*Jeffery et al., 2016*; *Yoder et al., 2019*). Head direction cells are found in the superficial and deep layers of dorsal presubiculum, also termed postsubiculum (*Boccara et al., 2010*), and most layer 3 neurons in the dorsal presubiculum are head direction cells (*Tukker et al., 2015*; *Preston-Ferrer et al., 2016*). These cells receive monosynaptic head direction inputs from the anterior thalamus (*Nassar et al., 2018*; *Peyrache et al., 2015*; *Balsamo et al., 2022*). Lesions of the PrS impair the visual landmark control of a cell's preferred direction in ATN and in LMN (*Goodridge and Taube, 1997*; *Yoder et al., 2015*). Presubicular lesions also induce place field instability in the hippocampus (*Calton et al., 2003*), suggesting this region may be crucial to the anchoring of directionally modulated neurons to environmental landmarks.

Landmark-based navigation depends on reliable visual cues. The PrS appears to receive direct projections from the primary visual cortex and indirect visual input via the retrosplenial cortex (*Van Groen and Wyss, 2003*; *Vogt and Miller, 1983*). The retrosplenial cortex encodes angular head velocity (*Alexander and Nitz, 2015*; *Keshavarzi et al., 2022*) and visual landmark information (*Auger et al., 2012*; *Clark et al., 2010*; *Sit and Goard, 2023*), and supports an array of cognitive functions, including memory, spatial and non-spatial context, and navigation (*Vann et al., 2009*; *Vedder et al., 2017*). In particular, some neurons in the dysgranular retrosplenial cortex encode landmark-dominated head direction signals (*Jacob et al., 2017*). Presubicular layer 3 cells receiving projections from both the anterior thalamus and the retrosplenial cortex (*Kononenko and Witter, 2012*) could update the compass and bind landmarks to head direction signals. Presubicular neurons could then broadcast integrated HD-landmark signals directly to the MEC, via deep layers to the ADN, and via layer 4 neurons to the LMN (*Huang et al., 2017*; *Yoder et al., 2015*; *Yoder and Taube, 2011*). The different target-specific presubicular projection neurons are well positioned to integrate anterior thalamic and retrosplenial inputs, but technical constraints have limited understanding of how known anatomical inputs are transformed into functional integrated output signals.

This work was therefore designed to examine the convergence of visual landmark information, that is possibly integrated in the RSC, and vestibular-based thalamic head direction signals, in the PrS. Retrograde tracing was used to confirm inputs to the PrS from the ATN and the RSC. The spatial distribution of ATN and RSC targets in the dorsal and ventral PrS was investigated by stereotaxic injection of viral vectors inducing anterograde expression of light-gated ion channels fused to fluorescent reporters. We found that superficial layers of the dorsal PrS are major targets of ATN and RSC projections. We analyzed functional convergence of these inputs in the PrS using dual-wavelength optogenetic stimulations in ex vivo brain slices, while recording from layer 3 and 4 pyramidal neurons. Both ATN and RSC projections made mono-synaptic excitatory connections with single layer 3 principal cells and mostly di-synaptic ones with layer 4 cells. We show that EPSPs induced with close to coincident timing by ATN and RSC fibers summed non-linearly in layer 3 neurons. Layer 4 cell firing is facilitated by cholinergic activation. These data provide insights into the integration of landmark and head direction inputs and their distribution to downstream targets by PrS pyramidal cells.

## Results

### ATN and RSC send strong axonal projections to the dorsal Presubiculum

The origins of the main afferents projecting to the presubiculum were explored by injecting retrogradely transported fluorescent beads into the PrS (*Figure 1A*). After 4 days, coronal slices were prepared to examine the injection site and transport of the beads. Some beads remained close to the PrS injection site, due in part to local projections. Strong bead signals were detected within two of the subnuclei that form the anterior thalamus, the anterodorsal (AD) and anteroventral (AV) part of ATN. Neurons in AD were labeled most strongly in the medial portion of the AD, while labeled neurons in AV were found in its lateral portion (*Figure 1B*). Many neurons in the RSC were labeled. Cells with somata in layers 2 and 5 of the dysgranular dRSC were labeled, while mostly layer 5 cells of granular gRSC contained beads (*Figure 1C*). Regions adjacent to the presubiculum, including the subiculum (Sub), parasubiculum (PaS), and the medial and lateral entorhinal cortices (MEC, LEC) were labeled as was the contralateral PrS. Beads were also detected in the laterodorsal thalamic nucleus (LD). *Figure 1—figure supplement 1* shows an example of a series of labeled coronal sections from one out of two mice, indicating some labeling also in the visual cortices, perirhinal cortex, the nucleus reuniens of the thalamus, the dorsolateral geniculate nucleus, and the claustrum. Some beads were observed in the CA1 region, which has not previously been reported. Potentially, though, this labeling could derive from a bead leak into the nearby PaS which is innervated by CA1 (*van Groen and Wyss, 1990a*). In summary, the ATN and RSC are the major sources of afferents projecting to the presubiculum with lesser inputs from other sites.

Projections from the ATN or RSC to the presubiculum were explored by injecting at these sites an anterogradely transported viral construct which expressed the modified Channelrhodopsin Chronos fused to GFP (*Figure 1D*). After 4 weeks, horizontal slices were prepared to verify the ATN injection site. Chronos-GFP labeling was mostly confined to the ATN, occasionally extending to medial thalamic and reticular nuclei nearby, possibly as bundles of projecting fibers (*Figure 1E*). After injection at the RSC site, coronal sections (*Figure 1F*) showed strong expression in layer 5, especially deeper zones, and layer 6 of both dysgranular and granular RSC, while labeled dendrites were evident in layer 1. Chronos-GFP expression extended throughout the RSC, from –1.7 to –3.7 posterior to Bregma on the antero-posterior axis.

Projections of the ATN to the presubiculum were examined in horizontal sections (*Figure 1G*). ATN axons expressing the Chronos-GFP construct targeted layers 1 and 3 of the presubiculum, mostly avoiding layer 2 (*Simonnet et al., 2017*; *Nassar et al., 2018*). Labeling was precisely limited to the presubiculum with a sharp reduction of green fluorescence at the posterior border with the parasubiculum, the anterior border with the subiculum, and the lateral border with deep layers. Axon terminals in layer 3 were not homogeneously distributed. Patches of higher density were evident in deep layer 3 (*Figure 1G*, left panel) or in the upper layer 3 (*Figure 1G*, fourth panel, and *Figure 1K*). Along the dorso-ventral axis, ATN axons projected from –2 to –4 mm ventral to Bregma. Fluorescence was not detected in adjacent regions, including the Sub, PaS, and DG. ATN axons avoided layer 2 of the PrS, including regions with patches of calbindin-positive neurons that project to contralateral PrS (*Preston-Ferrer et al., 2016*; *Figure 1K*).

RSC projections to the presubiculum (*Figure 1H*), traced with the fluorescent Chronos-GFP construct, also projected to layers 1 and 3. Unlike ATN axons, they were restricted to dorsal regions of the presubiculum (from –2 to –3 mm ventrally to Bregma on the dorso-ventral axis; *Figure 1H, I and J*). RSC axons also presented a patchy innervation pattern in L1-3, and tended to avoid microdomains of high-density ATN projections (*Figure 1G and K*; Figure 3). Overall, these data show ATN and RSC afferents innervate overlapping dorsal regions of the PrS, in layer 1 and parts of layer 3. Different spatial patterns of fiber terminations suggest distinct local connectivities.

### Layer 3 neurons are directly excited by both ATN and RSC afferents

We next examined physiological effects of ATN and RSC afferents on PrS pyramidal cells (*Rees et al., 2017*). Responses to photo-stimulation of the Chronos-GFP construct injected either in the thalamus or in the retrosplenial cortex (*Figure 2A*) were recorded in layer 3 neurons (*Figure 2B*). These cells had robust regular firing patterns in response to depolarizing current injections (*Figure 2C*). Intrinsic

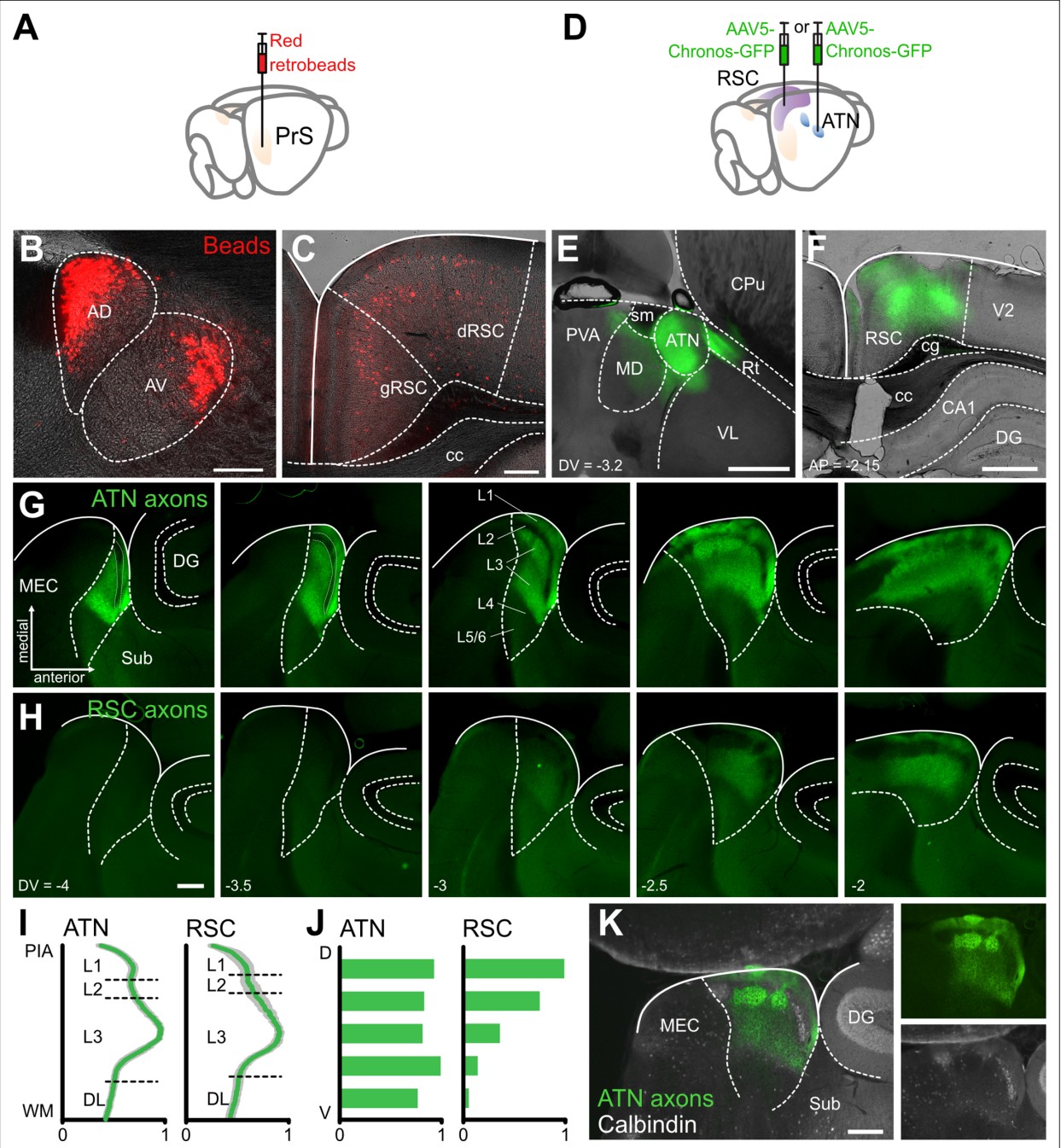

**Figure 1.** Anterior thalamic nuclei (ATN) and RSC send strong axonal projections to the superficial layers of dorsal presubiculum. (**A**) Retrograde labeling of cortical and subcortical regions projecting to the presubiculum (PrS) with retrobeads. (**B**) Ipsilateral anterodorsal and anteroventral thalamic nuclei are labeled with beads. (**C**) Ipsilateral granular (gRSC) and dysgranular (dRSC) retrosplenial cortex labeling. Scale bars (**B**, **C**), 200 μm. (**D**) Anterograde labeling of thalamic and retrosplenial projections to PrS with AAV-Chronos-GFP. (**E**, **F**) Injection sites in ATN (**E**) and RSC (**F**). AD: anterodorsal thalamic nucleus, AV: anteroventral thalamic nucleus, CA1: field of the hippocampus, cc: corpus callosum, cg: cingulum, CPu: caudate putamen, DG: dentate gyrus, MD: thalamic medial dorsal nucleus, MEC: medial entorhinal cortex, PaS: parasubiculum, PVA: paraventricular thalamic nucleus, anterior, Rt: thalamic reticular nucleus, sm: stria medullaris, Sub: subiculum, VL: thalamic ventrolateral nucleus. (**G**, **H**) Five sequential 100 μm horizontal slices of the parahippocampal region show ATN (**G**) and RSC (**H**) axons expressing Chronos-GFP (green). Dashed lines show limits of the parahippocampal region to the left and of the dentate gyrus to the right. Numbers show dorsoventral level with respect to bregma. Scale bars 100 μm. (**I**) Normalized profiles of fluorescent intensity for ATN and RSC projections to the presubiculum, from white matter (WM) to pia (PIA). Mean (green) ± SEM (gray), n=8 slices each. (**J**) Ventral (**V**) to dorsal (**D**) normalized distribution of ATN and RSC projections to the presubiculum, n=3 mice. (**K**) ATN axon labeling (green) is segregated from calbindin labeling (white) in the PrS. Scale bar, 100 μm. See also *Figure 1—figure supplement 1*.

*Figure 1 continued on next page*

*Figure 1 continued*

The online version of this article includes the following figure supplement(s) for figure 1:

**Figure supplement 1.** Brain regions providing input to the presubiculum.

**Figure supplement 2.** Calbindin immunostaining and GFP-expressing anterior thalamic nuclei (ATN) axons in horizontal sections.

properties of layer 3 cells were rather uniform and an unsupervised cluster analysis suggested they formed a single population (*Figure 2D*).

Blue light activation of either ATN or RSC axons expressing Chronos-GFP reliably evoked synaptic responses in whole-cell patch-clamp records from layer 3 cells. High-intensity stimulation (1–3 mW) evoked EPSCs or EPSPs with a probability of 84% (70/83 neurons from 14 mice) for ATN fibers and 85% (104/123 neurons from 20 mice) for RSC fibers (*Figure 2E*). Excitatory postsynaptic currents or potentials evoked by optical stimulation of thalamic and retrosplenial inputs will be referred to as oEPSCs and oEPSPs. Mean oEPSC amplitudes (*Figure 2F*) were similar for ATN (–305.0±61.0 pA, n=24) and RSC fiber stimuli (–305.8±59.1 pA, n=27). Response latency (*Figure 2G*) to ATN axon stimulation was 2.45±0.19 ms (n=24) and that for RSC axon stimulation was 2.93±0.23 ms (n=27). The variability in latency of oEPSCs (*Figure 2—figure supplement 1A, B*) induced by RSC afferent stimuli was 0.35±0.09 ms (n=27) and that for oEPSCs induced by ATN stimuli was 0.19±0.06 ms (n=24). These short oEPSC latencies suggest monosynaptic transmission. We confirmed this point by showing that oEPSCs were maintained in the presence of TTX (1 µM) and 4AP (100 µM) (*Figure 2H*, n=2 and 12 for ATN and RSC, respectively). EPSCs induced by stimulation of ATN and RSC fibers were reduced in amplitude and decayed more quickly in the presence of the NMDA receptor antagonist APV (50 µM) and were completely abolished by further application of the AMPA receptor antagonist NBQX (1 mM) (*Figure 2I*). The shape of oEPSCs induced by light stimulation of ATN and RSC fibers was comparable. Rise times were 1.38±0.15 ms for ATN EPSCs (n=15) and 1.38±0.08 ms for RSC-mediated EPSCs (n=15) and mean half-widths were 4.00±1.02 ms for ATN-induced ESPCs (n=15) and 4.34±1.18 ms for RSC EPSCs (n=15). The mean time constant (tau decay) for ATN-mediated EPSCs was 3.01±0.21 (n=15) and for RSC-induced EPSCs 3.17±0.27 (n=15) (*Figure 2—figure supplement 1*).

Optical stimulation of ATN and RSC fibers at lower light intensities elicited sub-threshold synaptic events in layer 3 cells. oEPSP amplitude for ATN-mediated events was 4.09±1.18 mV (n=11) and for RSC events it was 4.80±0.84 mV (n=11). Maximal rising slopes were 2.93±0.82 mV/ms for ATN-mediated events (n=11) and 3.02±0.53 mV/ms for RSC EPSPs (n=11), and decay time constants were 81.4±11.6 ms for ATN EPSPs (n=11) and 86.9±11.2 ms for RSC EPSPs (n=11). Rise times were shorter for subthreshold EPSPs induced by ATN afferents at 3.00±0.24 ms (n=11) while those for RSC-initiated EPSPs were 3.59±0.34 (n=11, $p$=0.042, Wilcoxon test; *Figure 2—figure supplement 1*). These data show that ATN and RSC axons make monosynaptic, glutamatergic excitatory contacts on layer 3 PrS cells.

The dynamics of responses to repetitive stimulation of layer 3 PrS cells were tested using 20 Hz trains of light stimuli to activate either ATN or RSC afferents. oEPSCs for both inputs followed depressing kinetics. Amplitude decreased for both inputs (*Figure 2J and K*), significantly after the fourth pulse (Friedman's test and Dunn's multiple comparison test). The *10/1* ratio was significantly lower than the paired-pulse ratio (PPR) in both cases (ATN *10/1* 0.56±0.05 vs PPR 0.86±0.05, n=15, $p$<0.0001, Wilcoxon test, RSC *10/1* 0.64±0.05 vs PPR 0.83±0.03, n=15, Wilcoxon test) (*Figure 2L*). ATN fiber stimulation induced PrS cell firing in current-clamp with a high probability for initial stimuli and lower stable probabilities for later stimuli in the train. Similarly, 20 Hz stimulation of RSC afferents evoked spikes with high probability for initial stimuli and decreasing probabilities for later stimuli (on fourth and $7^{th}$-$10^{th}$ pulses, Friedman's and Dunn's multiple comparison tests; *Figure 2M and N*). Responses to repeated subthreshold stimuli had similar dynamics (*Figure 2O*). Stimulation with lower light intensities (<1 mW) to initiate sub-threshold oEPSPs elicited different dynamic responses. ATN afferent trains elicited responses with depressing dynamics: the amplitude of the fifth oEPSP was significantly less than the first one (3.18±0.88 vs 5.11±1.64 mV, $p$=0.0185, Friedman's and Dunn's tests). In contrast, subthreshold 20 Hz stimulation of RSC afferents evoked little or no oEPSP depression (*Figure 2—figure supplement 1*). These data reveal distinct patterns of integration of repetitive activity in glutamatergic ATN and RSC afferents terminating on PrS layer 3 pyramidal cells.

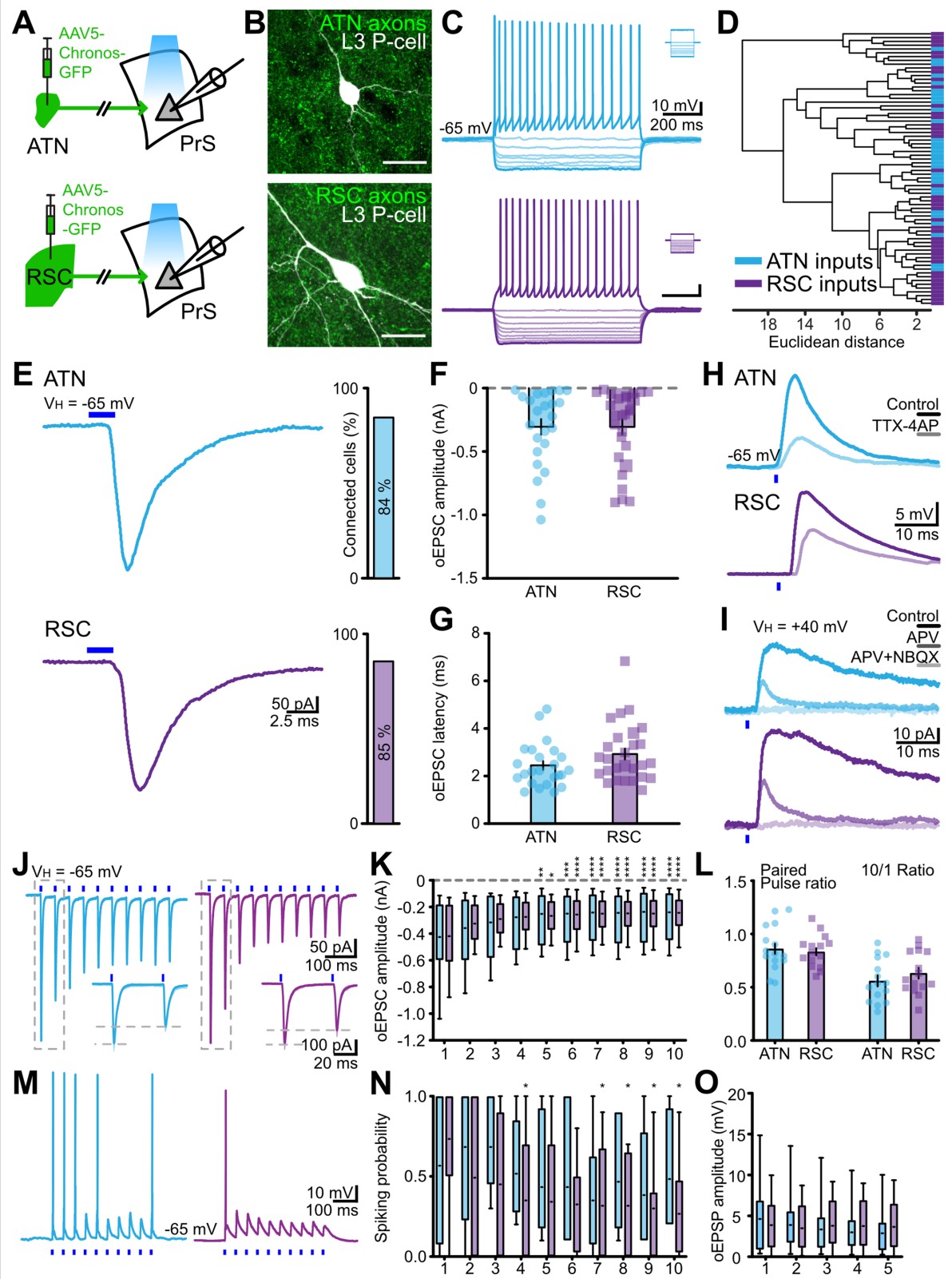

**Figure 2.** Responses of layer 3 presubicular cells to photoactivation of anterior thalamic nuclei (ATN) and RSC fibers. (**A**) Expression of AAV5-Chronos in ATN or RSC. (**B**) Biocytin-labeled layer 3 cells (white) and GFP-positive axons (green) from the ATN or RSC. Scale bar 10 μm. (**C**) Firing pattern of two layer 3 cells receiving inputs from the ATN (*cerulean*) and the RSC (*purple*). Insets show current commands. (**D**) Cluster analysis of physiological parameters for cells tested by stimulating ATN (n=29 cells /14 mice) or RSC (n=43 cells/20 mice) fibers. (**E**) Representative EPSCs evoked in layer 3 cells

*Figure 2 continued on next page*

*Figure 2 continued*

by light stimulation (blue bar) of ATN or RSC inputs. Right, the proportion of cells receiving ATN or RSC inputs. (**F**) Average amplitudes of ATN and RSC induced synaptic currents. Mann-Whitney U test revealed no significant difference (*p*=0.95). (**G**) Latency of EPSCs evoked by light stimulation of ATN (n=24 cells/12 mice) or RSC fibers (n=27 cells/14 mice). Mann-Whitney U test revealed no significant difference (*p*=0.13). (**H**) EPSPs induced in layer 3 cells (single traces) by stimulating ATN or RSC inputs in the absence and presence of 1 µM TTX and 100 µM 4-AP. (**I**) EPSCs induced by stimulating ATN or RSC fibers in the absence and presence of 100 µM APV and APV + 10 µM NBQX. Holding potential + 40 mV. (**J**) Voltage-clamp responses of layer 3 cells to 20 Hz train stimulations of ATN (*left*) and RSC (*right*) inputs. Insets show EPSCs in response to the first two stimuli. (**K**) oEPSC amplitudes for 10 ATN or RSC fiber stimuli at 20 Hz (ATN, n=15 cells/8 mice, RSC, n=15 cells/11 mice). Short middle line, mean; Min to max and quartiles. (**L**) Paired-pulse ratio (PPR) and 10/1 ratio (ratio between $10^{th}$ and fist event amplitudes) for ATN or RSC inputs. Wilcoxon matched-pairs signed rank test: ATN PPR vs 10/1 **** *p*<0.0001, RSC PPR vs 10/1 *** *p*=0.0001. (**M**) Current clamp traces showing action potentials and EPSPs evoked by 10 stimuli at 20 Hz. (**N**) Spiking probability during trains of 10 stimuli. ATN, n=6 cells/4 mice, RSC, n=12 cells/9 mice. Full line, median; short line, mean; Min to max and quartiles. In (**K** and **N**), **p*<0.05, ***p*<0.01, ****p*<0.001, *****p*<0.0001 from Friedman's test followed by Dunn's *post-hoc* test. (**O**) oEPSP amplitudes for trains of five stimuli at 20 Hz. See also .

The online version of this article includes the following source data and figure supplement(s) for figure 2:

**Source data 1.** Table showing the intrinsic electrophysiological physiological properties of layer 3 cells tested for responses to stimulating anterior thalamic nuclei (ATN) or RSC fibers.

**Figure supplement 1.** Comparison of evoked synaptic events in presubicular layer 3 neurons following photostimulations of anterior thalamic nuclei (ATN) or RSC axons.

## Convergence of ATN and RSC inputs on single layer 3 neurons

Are single layer 3 pyramidal cells innervated by both ATN and RSC afferents? It seemed likely since the probability of connection to a given cell was 84% for ATN fibers and 85% for RSC afferents, and cluster analysis (*Figure 2D*) provided no evidence for subpopulations of layer 3 cells in terms of intrinsic electrophysiological properties (see also *Balsamo et al., 2022*).

We tested the hypothesis using photostimulation at two different light frequencies to activate ATN and RSC afferents independently. However, this approach may be compromised since there is an overlap in the excitation spectra of blue-light activated Chronos (400–600 nm) and red-light activated Chrimson (450–700 nm ; *Klapoetke et al., 2014*). Excitation of Chronos with high intensities of blue light might also excite Chrimson. Stimulating with red light at 630 nm should elicit neurotransmitter release in Chrimson-containing but not Chronos-expressing fibers, which we found was indeed the case. Blue light stimuli at 470 nm and intensities higher than 0.01 mW elicited oEPSPs in layer 3 cells of animals with Chronos-expressing fibers. In contrast, blue light intensities higher than 0.25 mW were needed to induce oEPSPs in different animals with Chrimson-expressing axons (*Figure 3—figure supplement 1A–D*). Moreover, the amplitude of these events was smaller in Chrimson-injected than in Chronos-injected mice. Thus, in our experimental conditions, blue light intensities up to 0.25 mW could be used to excite Chronos-positive axons with confidence that Chrimson-expressing fibers would not be stimulated.

We also tested the inverse, red light component of the dual stimulus strategy. Intensities were tested in mice injected either with AAV5-Chronos to label ATN fibers or AAV5-Chrimson to label RSC afferents. The strategy was validated by showing that red light induced oEPSPs in Chrimson-containing, but not Chronos-expressing fibers, while blue light evoked synaptic events in Chronos-, but not Chrimson-expressing fibers (*Figure 3—figure supplement 1E–K*).

Injection in the same animal of the Chronos construction to the thalamus and that for Chrimson to the retrosplenial cortex let us visualize ATN and RSC afferents to the presubiculum (*Figure 3A, B*). Axon terminal fields of ATN and RSC projections were evident in superficial layers of dorsal PrS (*Figure 1*). Layer specificities and labeling inhomogeneities were apparent, as were high-density ATN patches below layer 2 with no RSC axon innervation (*Figure 3B*). Layer 3 pyramidal cells could be innervated by ATN and RSC contacts both with apical dendrites in layer 1 and basilar dendrites of layer 3 (*Figure 3D*). We found optical stimulation of ATN and RSC axons evoked synaptic responses in 14 out of 17 layer 3 pyramidal cells tested (current-clamp and voltage-clamp; *Figure 3E, F*).

In order to quantify substrates of these innervations, we counted the numbers of labeled ATN or RSC terminals located at less than 1 µm from dendrites of biocytin-filled layer 3 pyramidal cells (n=6; *Figure 3G*). The distribution of these potential contact sites on dendritic trees of layer 3 neurons was diverse. Some neurons exhibited highly segregated domains receiving either ATN or RSC inputs. For instance, cell 1 had a high proportion of clustered RSC putative synapses on its apical tuft while

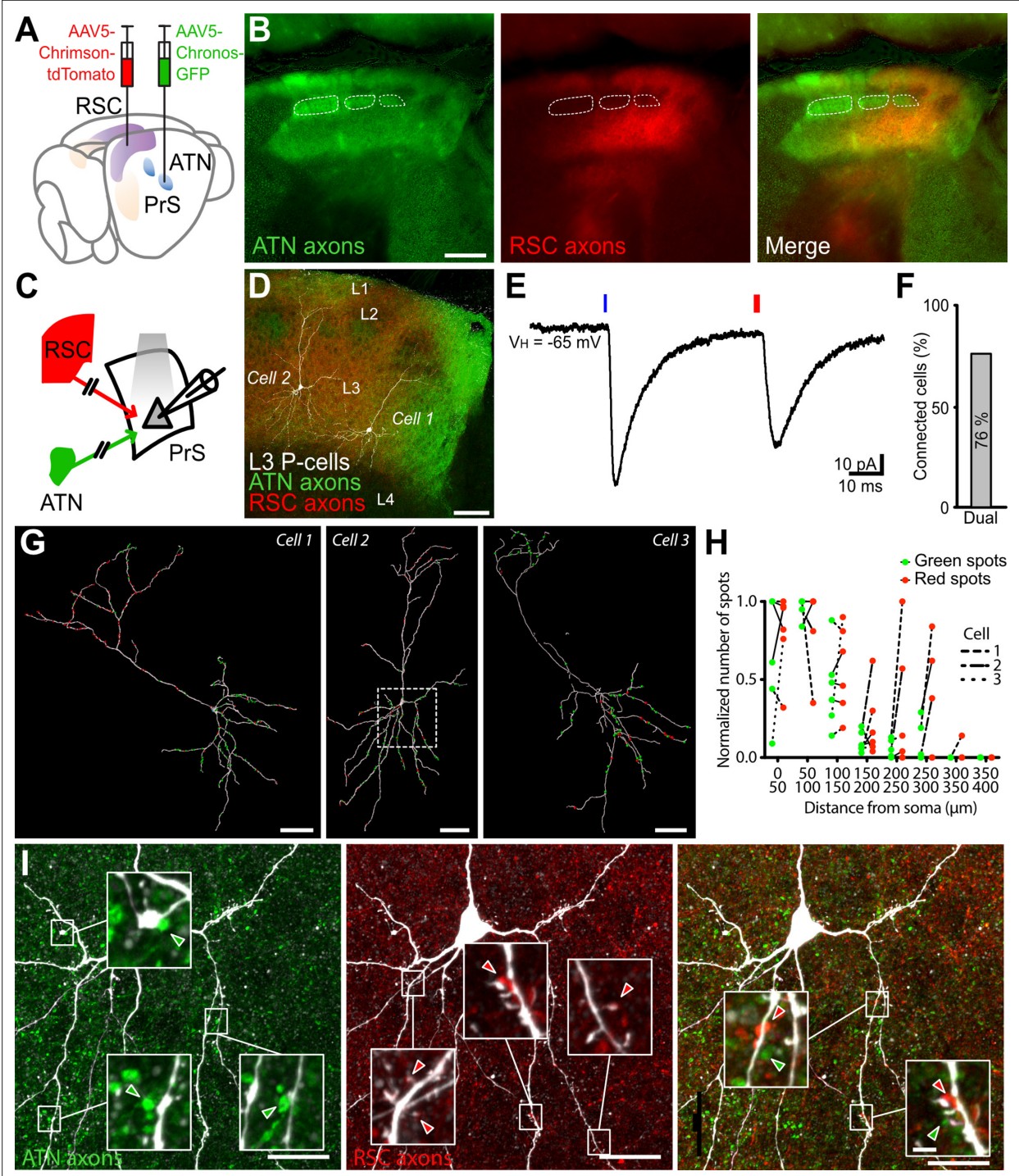

**Figure 3.** Anterior thalamic nuclei (ATN) and RSC axons converge in dorsal presubiculum and contact single layer 3 pyramidal neurons. (**A**) Expression of the blue light-sensitive opsin Chronos in ATN and the red light-sensitive opsin Chrimson in RSC. (**B**) Axons from ATN (GFP labeled, green) and RSC (tdTomato labeled, red) overlap in layer 1 and 3 of dorsal presubiculum. Layer 2 possesses patches of axon-dense and axon-poor zones. RSC fibers avoid axon-dense microstructures formed by ATN fibers in upper layer 3. Scale bar, 200 μm. (**C**) Independent dual-wavelength optogenetic stimulation of light-sensitive afferent fibers in presubicular slices. (**D**) Two biocytin-labeled PrS layer 3 pyramidal cells surrounded by ATN (green) and RSC (red) axons. (**E**) Patch clamp recording from a layer 3 neuron shows optical EPSCs following photostimulation of ATN axons (blue light) and RSC axons (red light). (**F**) 76% of layer 3 pyramidal neurons tested (n=17 cells/4 mice) received both ATN and RSC input. (**G**) Distributions of putative synaptic contacts from ATN (green) and RSC (red) on the dendrites of three layer 3 neurons. Scale bar 50 μm. The boxed area on Cell 2 is shown in panel **I**. (**H**) Normalized number of green and red spots for 6 neurons as a function of the distance from soma. Paired values are indicated by dotted lines for the 3 cells in **G**. (**I**)

*Figure 3 continued on next page*

*Figure 3 continued*

Examples of ATN-labeled (left), RSC-labeled (middle), and both (right) synapses closely apposed to dendrites of a biocytin-filled layer 3 pyramidal cell. Scale bar 20 μm. Insets show representative high-magnification images. Scale bar, 2 μm. See also *Figure 3—figure supplement 1*.

The online version of this article includes the following figure supplement(s) for figure 3:

**Figure supplement 1.** Calibration of blue and red light stimulation of Chronos and Chrimson.

basilar dendrites were mostly surrounded by potential thalamic terminals. In cell 2, RSC terminals were clustered close to apical dendrites and also together with ATN terminals close to basilar dendrites. In contrast, many ATN terminals were located closer to apical dendrites of cell 3. Segregation of cortical RSC inputs to apical dendrites and thalamic ATN inputs to basal dendrites might favor supralinear EPSP summation. However, afferent terminal distributions differed, and we found no clear pattern of sub- or supralinear summation (cell 1, supralinear summation (Dual/summed EPSP, 1.39); cell 2, roughly linear (0.91); cell 3, sublinear (0.69), *Figure 3G, H*). Furthermore, we noted certain potential contacts were located with separations less than 20 μm on the same dendrite (*Figure 3I*). Such proximity might underly local dendritic computations and supra-linearities, such as NMDA-mediated dendritic spikes which are suggested to contribute to non-linear summation of synaptic events (*Larkum et al., 1999*; *Larkum et al., 2007*; *Larkum et al., 2009*).

## Supralinear integration and amplification of ATN and RSC excitatory postsynaptic potentials

We explored interactions between excitatory inputs to layer 3 neurons by stimulating ATN and RSC axons separately with blue or red light. Excitatory synaptic potentials evoked by ATN or RSC axon stimulation were compared to responses elicited when both sets of axons were activated at short time intervals (dual; *Figure 4A*). The mean amplitude of dual oEPSPs (synchronous light onset, *Figure 4A–C*) was significantly larger than calculated summed amplitudes of single oEPSPs (n=11 cells, dual/summed amplitude 2.06±0.55 mV, $p=0.0068$, Wilcoxon test). Charge transfer, quantified as the surface under the dual oEPSP, was greater than the summed values for single oEPSPs (Dual/summed surface 2.21±0.54 mV.s, n=11, $p=0.0098$, Wilcoxon test). These mean values cover variability between individual neurons. The summation of responses to ATN and RSC axon stimuli was supralinear in 7 out of 11 cells tested, linear for 3 cells and sublinear for 1 cell.

Dynamic responses to repetitive stimulations were also transformed by dual photostimulation. Amplitudes of 5 dual oEPSPs elicited at 20 Hz were higher than for a linear summation (Amplitudes in mV, Summed vs. Dual: Stim 1, 7.7±1.9 vs 11.1±1.9; Stim 2, 6.7±1.6 vs 9.5±1.5; Stim 3, 6.5±1.4 vs 9.5±1.5; Stim 4, 6.1±1.3 vs 9.2±1.5; Stim 5, 5.9±1.3 vs 8.6±1.4, $p=0.0117$, Two-way ANOVA; *Figure 4E*) and charge transfer, quantified as the surface under the 5 oEPSPs, also typically summed supralinearly (20 Hz, Surface in mV.s, Summed vs. Dual: Stim 1, 136.9±30.6 vs 221.2±41.5; Stim 2, 143.3±25.7 vs 242±43.9; Stim 3, 147.3±22.5 vs 259.2±44; Stim 4, 145.1±22 vs 263.5±49.2; Stim 5, 145.6±23.2 vs 258.7±45.2, $p=0.0128$, two-way ANOVA; *Figure 4F*). Trains of dual ATN and RSC fiber stimulations also showed increased oEPSP integrals, compared to the stimulation of either one set of fibers alone, indicating stronger excitation over time (*Figure 4E and F*). Taken together, these data show supralinear summation of ATN and RSC inputs in most single layer 3 pyramidal cells, and also a facilitation of dynamic responses to repetitive inputs at 20 Hz.

We next asked how efficiently action potentials were induced by combined ATN and RSC inputs. Firing probability of layer 3 cells was higher for dual than for single input stimulation at a membrane potential of –65 mV (ATN 0.008 vs Dual 0.268, n=5, $p=0.0216$, RSC 0.024 vs Dual 0.268, n=5, $p=0.1195$, Friedman's and Dunn's tests) and higher still at –55 mV (*Figure 4G and H*). Firing was induced over a narrow window of time delays between stimulation of the two inputs (*Figure 4I and J*). It occurred for delays of –2 to +5 ms (RSC preceding ATN stimulus) reaching a maximum at a delay of +1 ms.

Supralinear summed responses of excitatory synaptic events may be mediated via the activation of NMDA receptors or voltage-dependent intrinsic currents that amplify EPSPs (*Fricker and Miles, 2000*), especially if synaptic inhibition is reduced. We assessed these mechanisms in records made from layer 3 pyramidal neurons using a cesium-based internal solution to favor cationic amplifying currents, and also containing QX314 to suppress action potentials (*Figure 5*). In 3 cells (from 2 mice), photostimulation of ATN fibers evoked oEPSPs of amplitude 10.9±2.0 mV and RSC fiber stimulation

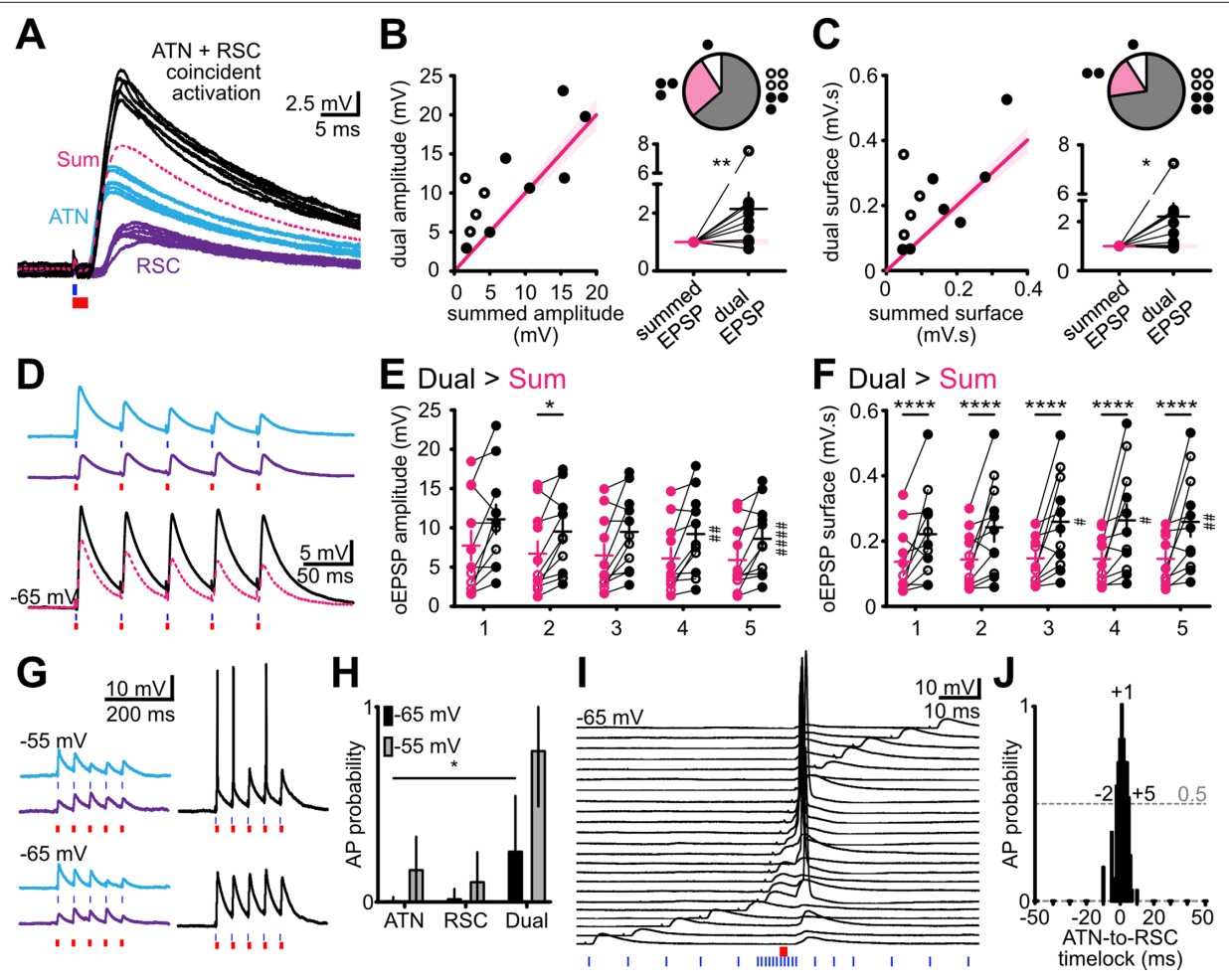

**Figure 4.** Supralinear summation of EPSPs and action potential firing following photostimulation of anterior thalamic nuclei (ATN) and RSC axons. (**A**) Optical EPSPs in layer 3 neurons in response to blue light activation of ATN axons (cerulean traces), red light activation of RSC axons (purple traces). Supralinear summation of EPSPs following coincident activation of both ATN and RSC axons (black traces). The pink broken line indicates the calculated linear sum of oEPSPs evoked by stimulation of either ATN or RSC axons. Records from cell 1, *Figure 3D*. (**B**) Amplitudes of dual ATN and RSC oEPSPs plotted as a function of the sum of separate ATN and RSC stimulations (*left*). Each circle is a cell (n=11 cells/4 mice). Pink line (±10%) indicates linearity. Pie charts give the number of tested layer 3 neurons with supralinear (gray), linear (pink), or sublinear (white) summation. oEPSPs normalized to linear sum (*bottom right*). Solid circle, Chronos in ATN/Chrimson in RSC, empty circle, Chronos in RSC/Chrimson in ATN. *p<0.05, **p<0.01, from Wilcoxon test. (**C**) As in (**B**) for dual ATN and RSC oEPSP integrals. (**D**) oEPSPs induced by 20 Hz stimulation of ATN (cerulean), RSC (purple), or both (black). The pink broken line indicates the calculated linear sum of oEPSPs evoked by separate ATN and RSC photostimulation. (**E**) Amplitudes of dual oEPSPs compared to those of the sum of ATN and RSC oEPSPs show that supralinearity increases across five stimulations. Two-way ANOVA, p=0.0117. *p<0.05, ****p<0.0001, Šidák's multiple comparison test. # p<0.05, ## p<0.01, Friedman's and *post-hoc* Dunn's test. (**F**) As in **E**, for dual ATN and RSC oEPSP integrals. Two-way ANOVA, p=0.0128. (**G**) Excitatory postsynaptic responses to photostimulation of ATN (blue light, cerulean traces) or RSC axons (red light, purple traces), or both (blue and red light, black traces) at resting membrane potential (–65 mV) or at a depolarized holding potential (–55 mV). Synaptic excitation led to action potentials when dual ATN and RSC stimuli reached firing threshold. (**H**) Action potential (AP) probability for either or both stimuli at –65 and –55 mV. Data are presented as mean ± SEM. (**I**) Action potentials were induced in presubicular layer 3 neurons by near-coincident activation of ATN axons (red light) and RSC axons (blue light). Time delays varied from –50 to +50 ms. (**J**) Firing probability was maximal for short delays between –2 to +5 ms (RSC preceding ATN).

induced events of 13.7±2.5 mV. The response dynamics to 5 light pulses at 20 Hz were moderately facilitating (second vs first amplitude ratio: ATN 1.39, RSC 1.17; fifth vs first: ATN 1.18, RSC 1.20; n=3). Coincident activation of ATN and RSC afferents induced supralinear oEPSP summation. Furthermore, repetitive stimulation generated large all-or-none depolarizations on the second or third stimulus at 20 Hz (second vs first amplitude ratio: 1.52; fifth vs first: 2.90; n=3 cells, *Figure 5A*). It may be mediated by VGCC or a QX-314-resistant Na + inward current (*Fricker et al., 2009*). This component appeared with higher probability and shorter latency in the presence of the GABA$_A$ receptor blocker gabazine

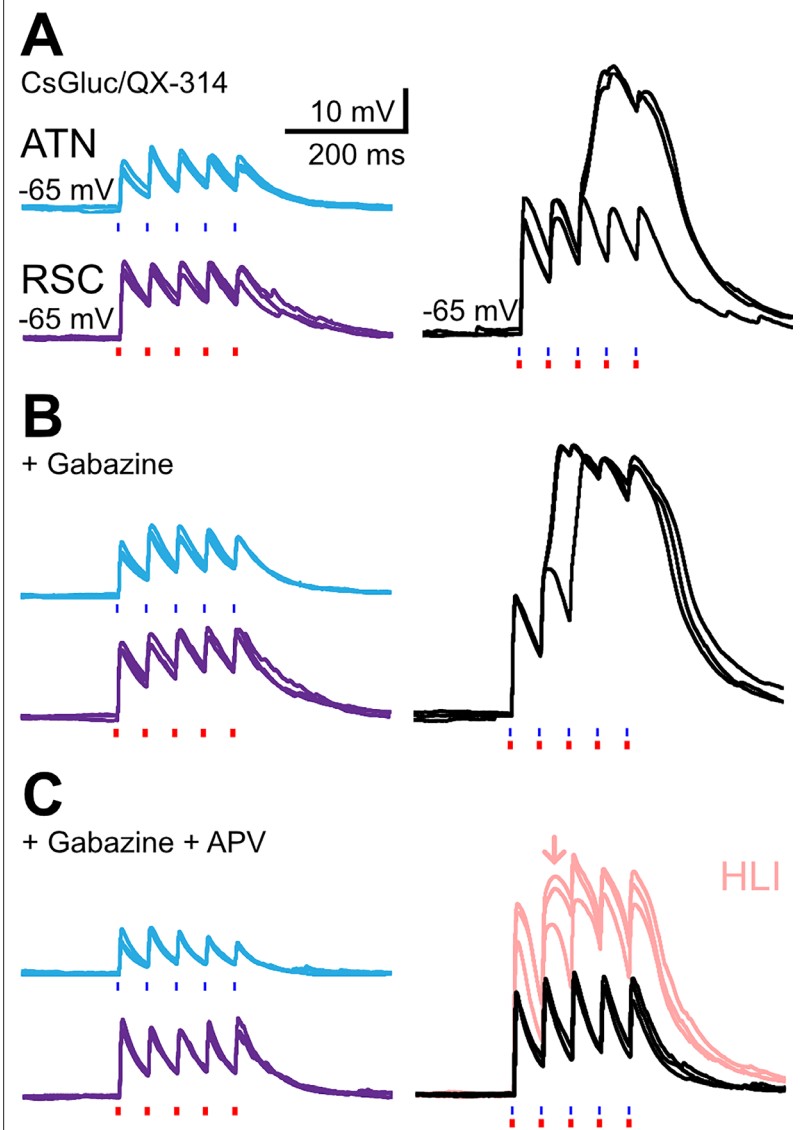

**Figure 5.** EPSP amplifications in layer 3 pyramidal neurons. (**A**) Optical EPSPs evoked in layer 3 pyramidal neurons following the stimulation of anterior thalamic nuclei (ATN) axons (blue light, cerulean traces), RSC axons (red light, purple traces), or both (blue and red light, black traces). The recording pipette contained a cesium gluconate-based internal solution and the Na⁺ channel blocker QX-314. A large all–or-none EPSP amplification occurred for dual stimuli at 20 Hz, on some trials. (**B**) In the presence of the GABA$_A$ receptor antagonist gabazine (10 μM), dual EPSP amplification occurred earlier in the train. (**C**) The additional presence of the NMDA receptor antagonist APV (100 μM) abolished dual EPSP amplification (black trace). EPSP amplification was partially restored by increasing red light intensity 2x (pale pink traces).

(*Figure 5B*). The NMDA receptor antagonist APV largely abolished the dual EPSP amplification initially (*Figure 5C*, black trace; second vs first integral ratio: 1.22; fifth vs first: 1.32), although amplification was partially restored by increasing RSC stimulus intensity (pale pink trace; 'amplification' refers to the shape of the plateau-like prolongation of the peak, most pronounced on the second EPSP, indicated with an arrow). NMDA receptor activation thus assists depolarization towards the threshold of a voltage-dependent process contributing to supra-linear EPSP summation but is not the only charge carrier involved.

## Cholinergic modulation and recruitment of presubicular layer 4 neurons by ATN and RSC afferents

Presubicular layer 4 neurons are intrinsic bursting pyramidal neurons that project to the lateral mammillary nucleus (*Huang et al., 2017*). This pathway is critical for the coordinated stable landmark control of HD cells in the thalamus and throughout the HD cell circuit (*Yoder et al., 2015*; *Yoder et al., 2017*). To investigate the input connectivity of layer 4 principal cells, we recorded responses of these neurons to stimulation of ATN and RSC afferents.

Layer 4 neurons labeled by retrograde tracers injected in the lateral mammillary nucleus were located in the *lamina dissecans* of the presubiculum, below layer 3, where thalamic axons ramify (*Figure 6A–D*). At the proximal end of the presubiculum, and in its continuity (*Ishihara and Fukuda, 2016*), some subicular neurons projecting to the medial portion of the mammillary bodies are also labeled (*Figure 6B*). The apical dendrites of layer 4 presubicular neurons extended towards presubicular layer 1 as previously described (*Huang et al., 2017*), but tended to circumvent layer 3 and avoid thalamic afferent, by swerving towards the subiculum. Apical dendrites of some neurons crossed layer 3 obliquely, while others avoided thalamic afferents in layer 3 (*Figure 6C and D*, *Figure 6—figure supplement 1*).

Layer 4 neurons had a more depolarized resting membrane potential, lower input resistance, and time constant than layer 3 neurons

(*Figure 6—figure supplement 2* compares active and passive properties). A characteristic voltage sag in responses to hyperpolarizing steps indicated the presence of an I$_h$ current. Layer 4 neurons discharged bursts of two or three action potentials at the onset of a depolarizing step current injection and also after the offset of hyperpolarizing steps (*Figure 6E*). These bursts were abolished by the T-type Ca²⁺ channel blocker TTA-P2 (*Figure 6F*).

We next recorded responses of layer 4 neurons to optical activation of ATN or RSC afferents together with effects of the same stimuli on layer 3 pyramidal cells (*Figure 6G*). Overall, latencies

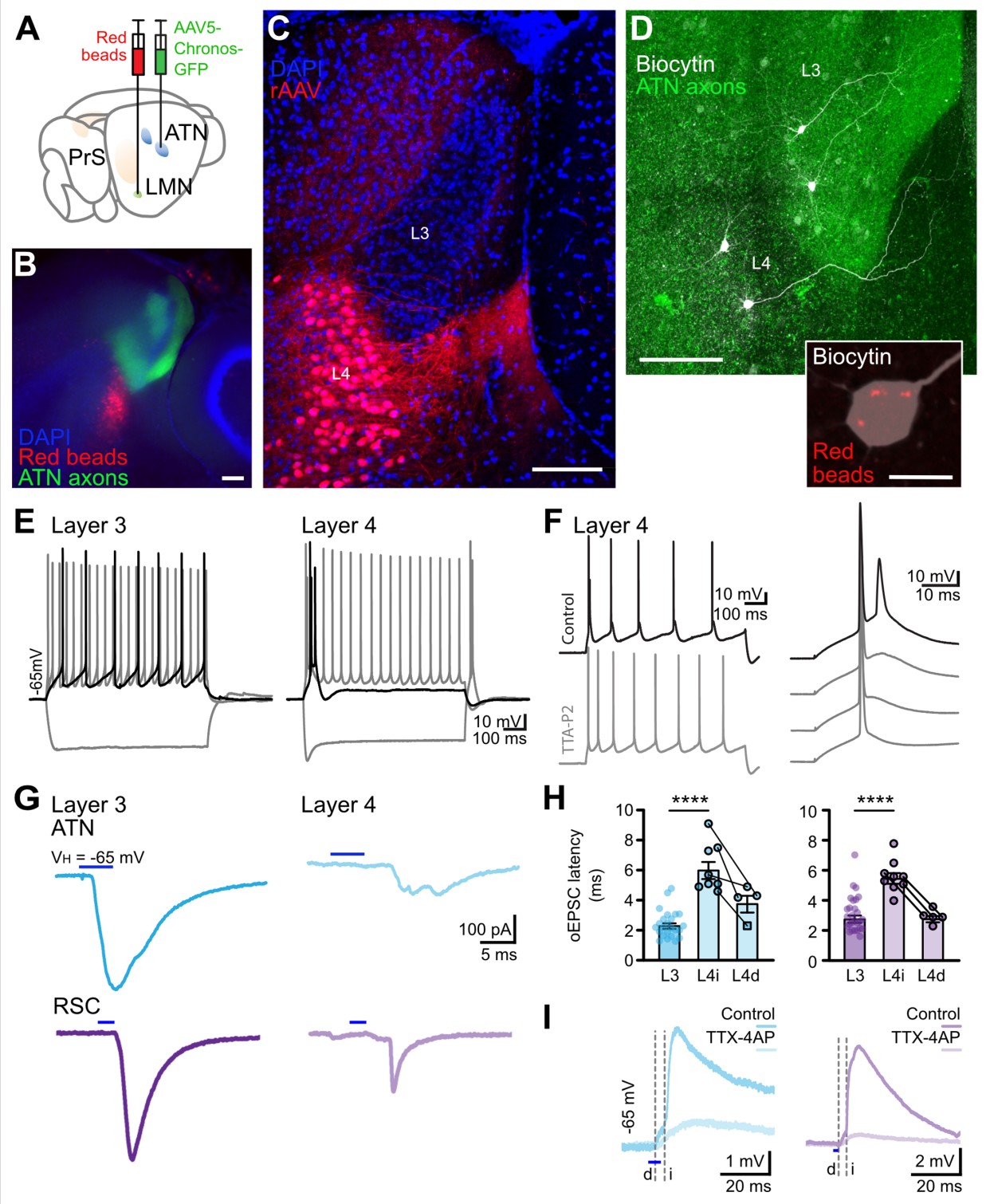

**Figure 6.** Presubicular lateral mammillary nucleus (LMN)-projecting layer 4 neurons avoid thalamo-recipient layer 3 and receive little direct input from anterior thalamic nuclei (ATN) and RSC. (**A**) Expression of Chronos-GFP in ATN and retrograde labeling of neurons that target LMN. (**B**) Thalamic axons (green) in superficial layers 1 and 3 of presubiculum. Retrobeads label cell bodies of presubicular layer 4 cells (red). (**C**) Retrograde rAAV2-tdTomato label cell bodies and dendrites of layer 4 LMN projecting neurons (red). Apical dendrites of layer 4 pyramidal neurons avoid layer 3 where thalamic axons ramify. (**D**) Presubicular slice containing two layer 3 and two layer 4 neurons filled with biocytin (white) and GFP-expressing thalamic axons (green). Scale bar 100 μm. Inset, retrobeads (red) in the soma of a biocytin-filled LMN-projecting layer 4 neuron. Scale bar, 10 μm. (**E**) Layer 3 neurons are regular spiking and layer 4 neurons are burst firing, initially and at rebound, in response to current injection. Black trace, rheobase. (**F**) T-type Ca$^{2+}$ channel

*Figure 6 continued on next page*

*Figure 6 continued*

blocker TTA-P2 (1 µM) suppressed burst firing in presubicular layer 4 neurons, while single action potentials were preserved. (**G**) Representative oEPSCs in layer 3 (left) and layer 4 (right) pyramidal cells, in response to stimulation of ATN (cerulean) or RSC (purple) inputs. (**H**) oEPSC latencies in layer 3 and layer 4 cells, for ATN inputs (left, cerulean), or RSC inputs (right, purple). Each dot is a cell. Same layer 4 cells are indicated by connecting lines to show the difference in latency for direct and indirect synaptic responses. (**I**) oEPSPs in layer 4 neurons in response to stimulation of ATN (cerulean) or RSC (purple) inputs in control and in the presence of TTX (1 µM) and 4-AP (100 µM). Dashed lines indicate the timing of the large disynaptic component of the responses (i, indirect), and the small monosynaptic response (d, direct), isolated in TTX-4AP. See also *Figure 6—figure supplement 1*.

The online version of this article includes the following figure supplement(s) for figure 6:

**Figure supplement 1.** Apical dendrites of presubicular layer 4 neurons avoid the thalamorecipient layer 3.

**Figure supplement 2.** Electrophysiological passive and active intrinsic properties of layer 3 vs. layer 4 neurons.

of oEPSCs in layer 4 neurons were longer than for layer 3 (ATN layer 4, 6.2±0.6 ms, n=8, vs. layer 3, 2.4±0.2 ms, n=24; RSC layer 4, 5.6±0.4 ms, n=9, vs. layer 3, 2.9±0.2 ms, n=27; *Figure 6H*), indicating possible polysynaptic excitation of layer 4 neurons. Bath application of TTX-4AP did not abolish oEPSPs entirely, leaving a low amplitude component with short, potentially monosynaptic latencies (latency ATN 3.9±0.6 ms, RSC 2.9±0.2 ms, n=5; *Figure 6I*).

Comparison of the timing of synaptic events and firing (*Figure 7A*) showed that oEPSP onset in layer 4 neurons occurred after firing in the layer 3 neuron, following ATN afferent stimuli, in 4 out of 5 cell pairs. We also observed this sequence when RSC input was activated, in one tested pair. Depolarization of the layer 3 cell via the patch pipette, initiated firing, but EPSPs were not elicited in any simultaneously recorded layer 4 neuron (0 out of six cell pairs tested). These data suggest that excitation of layer 3 cells by ATN and likely by RSC afferents is transmitted to layer 4 neurons, even in the absence of direct evidence for mono-synaptic coupling between cell pairs.

In records from pairs of layer 3 and 4 neurons (n=3; *Figure 7D and E*) layer 3 cells responded with precisely timed action potentials to low intensity stimulation of ATN afferents, while only very small oEPSPs were initiated in layer 4 neurons. Increasing the excitatory drive by stimulating RSC afferents elicited larger oEPSPs more reliably in layer 4 neurons. Higher intensity stimulation of both ATN and RSC axons could evoke bursts of action potentials, with the activation of an underlying $Ca^{2+}$ current in layer 4 cells (*Figure 7E*). We hypothesize that increasing the activity of layer 3 neurons by strong (and non-specific) stimulation of ATN and RSC afferents is needed to induce discharges in layer 4 cells. We should also note that strong photostimulation of a single set of afferent fibers could initiate discharges in layer 4 pyramidal cells, as was shown in records from mice where only one afferent brain area, ATN or RSC, expressed an opsin (*Figure 7—figure supplement 1*).

Overall, it seems probable that layer 3 neurons relay activity projected by ATN and RSC inputs onto layer 4 pyramidal neurons. However, evidence on this point could be improved. In n=9 paired recordings, we did not detect functional synapses between layer 3 and layer 4 neurons. Anatomically, putative synaptic contacts between filled axons of layer 3 and dendrites of layer 4 cells were not evident (n=5 filled layer 4 cells; see also *Peng et al., 2017*). We did find a possible synaptic connection between two neighboring L4 neurons in one case (*Figure 6—figure supplement 1D*), with a close apposition between axon and dendrite. Recurrent excitation may promote bursting in a positive feedback loop in layer 4 neurons, and NMDA receptor-related EPSP amplification favors burst firing (*Figure 7F*).

Neuromodulatory factors may help drive layer 4 cells to fire action potentials. As the presubiculum is rich in acetylcholinesterase (*Slomianka and Geneser, 1991*), we examined excitability changes in the presence of a broad cholinergic agonist. The application of Carbachol (10 µM; *Figure 7G*) led to a depolarization of the recorded layer 4 neurons' membrane potential by 13±4 mV, with increased action potential firing during a positive step current injection, and from the baseline (*Figure 7Gi,ii*).

## Discussion

Understanding the anchoring of HD signals to environmental landmarks provides a useful framework to ask how external information is integrated to revise and improve brain representations. Here, we show that thalamic ATN afferents signaling HD data and cortical RSC projections that carry landmark information converge on layer 3 pyramidal cells of the dorsal presubiculum. Independent optogenetic stimulation of these two glutamatergic afferents in slices was used to define mechanisms of their

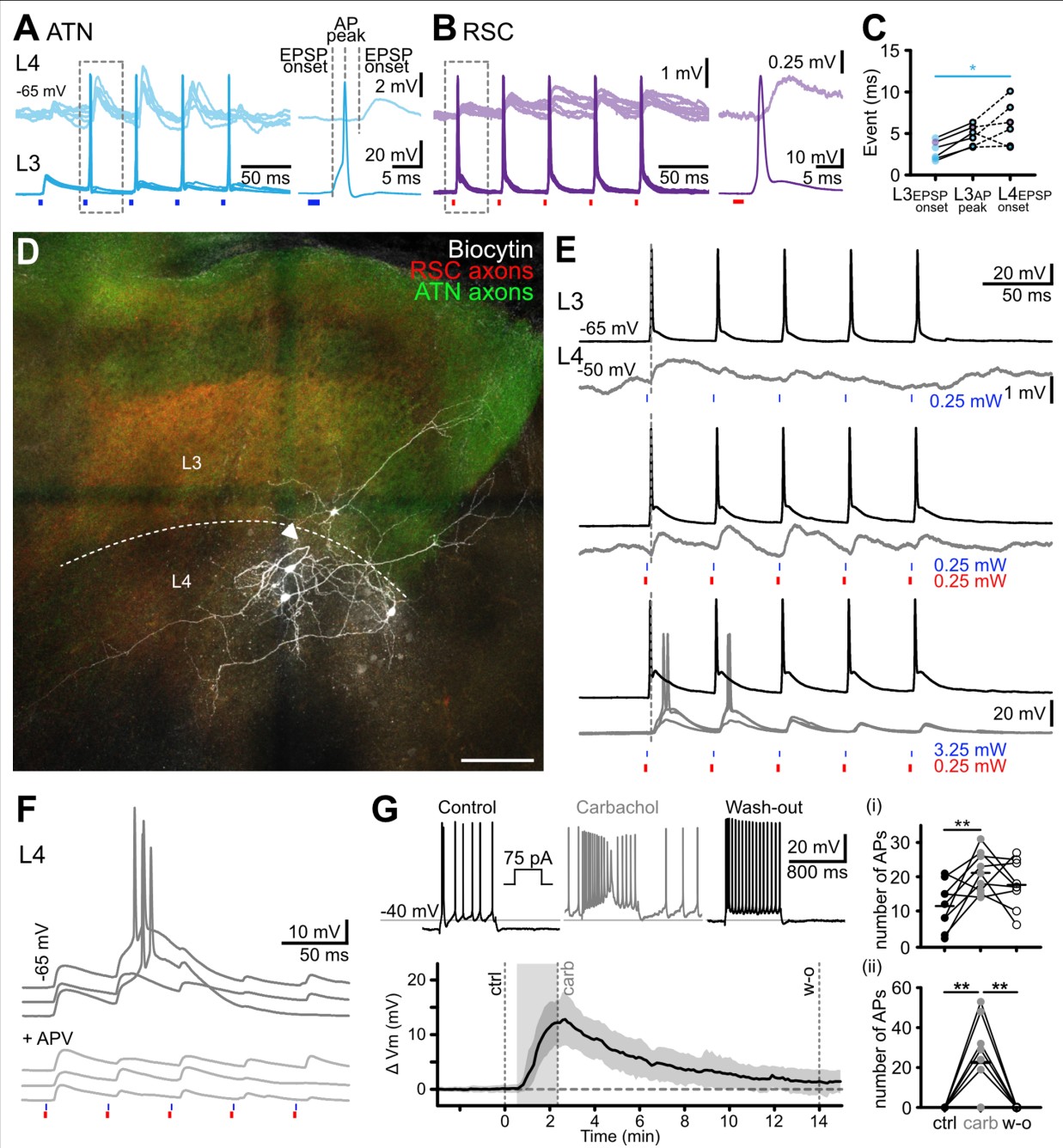

**Figure 7.** Cross-laminar activation of lateral mammillary nucleus (LMN)-projecting layer 4 bursting neurons. (**A**) Simultaneous records of a layer 4 and a layer 3 neuron during photostimulation of anterior thalamic nuclei (ATN) afferents. EPSP onset was delayed in the layer 4 neuron. Right panel, expanded view of the boxed area. (**B**) Simultaneous records of a layer 4 and a layer 3 neuron during photostimulation of RSC afferents. As in **A**, layer 4 neurons responded with a delay. Right panel, expanded view of the boxed area. (**C**) Latencies of synaptic activation indicated by the dotted lines in **A**, **B** (ATN stimulation, n=4 cells; RSC stimulation, n=1). EPSP onset and antero-posterior (AP) peak from layer 3 neurons (n=5). Dotted lines link to the EPSP onset in simultaneously recorded layer 4 neurons (n=5). $p<0.05$, Kruskal-Wallis multiple comparison test. (**D**) Biocytin-labeled layer 3 and layer 4 pyramidal neurons, in a presubicular slice containing thalamic (green) and retrosplenial (red) axons. The apical dendrite of one layer 4 neuron makes a U-turn (arrowhead), away from layer 3, where the thalamic axons ramify. The neighboring layer 4 neuron's apical dendrite crosses the thalamo-recipient layer 3 for a short distance before arborizing outside of ATN targeted area, towards the subiculum, on the right. Another biocytin-filled layer 4 neuron's dendrites extend toward the deep layers. Scale bar, 100 μm. (**E**) Simultaneous records from a layer 3 and a layer 4 cell to ATN input stimulation (top, 0.25 mW, blue light), ATN and RSC input stimulation (middle, 0.25 mW blue and 0.25 mW red light; light intensities compatible with independent photostimulation) or non-specific ATN and RSC input stimulation (bottom, 3.25 mW blue and 0.25 mW red light). ATN fibers expressed Chronos-GFP (green) and RSC fibers expressed Chrimson-tdTomato (red). (**F**) Top, oEPSPs and bursts of action potentials in a layer 4 neuron, evoked by dual

*Figure 7 continued*

wavelength stimulation of ATN and RSC afferents at 20 Hz. Amplifications of dual oEPSPs led to firing. Bottom, the NMDA receptor antagonist APV (100 μM) reduced EPSP amplification and prevented action potential firing. (**G**) Layer 4 bursting neurons are sensitive to the acetylcholine receptor agonist carbachol (10 μM). Action potential firing in response to step current injections in control (black), in the presence of carbachol (gray), and after wash-out (black). Bottom graph, membrane potential depolarization during a 2 min carbachol application. The number of action potentials increased during the depolarizing steps (i) and on the baseline (ii). See also *Figure 7—figure supplement 1*.

The online version of this article includes the following figure supplement(s) for figure 7:

**Figure supplement 1.** Regular firing layer 3 vs. intrinsically bursting layer 4 neurons responded to high-intensity light stimulations of anterior thalamic nuclei (ATN) or RSC afferents.

**Figure supplement 2.** Schematic models for landmark anchoring of head direction (HD) signals.

integration in the presubiculum. Most layer 3 cells were innervated by both ATN and RSC fibers, sometimes close on the same dendritic branches, sometimes on different dendrites. Nearly coincident EPSPs (2–5 ms) induced independently by ATN and RSC afferents evoked non-linear membrane responses to trigger layer 3 cell firing, transmitted to the MEC. In a second processing step, layer 3 neurons also excite layer 4 pyramidal cells which receive little direct ATN and RSC innervation. These burst-firing neurons may project a distinct, di-synaptically mediated, visually updated HD signal to the LMN. Our data thus suggest that the dorsal presubiculum integrates HD and landmark signals producing two distinct output signals which are transmitted to different regions. Cholinergic modulation may facilitate responses to salient stimuli for flexible anchoring to landmarks.

## Anatomical convergence of ATN and RSC projections in the dorsal presubiculum

Retrograde tracing in this study confirmed strong projections to the presubiculum from the ATN and the RSC to the presubiculum (*Figure 1*). Anterior thalamic fibers (*van Groen and Wyss, 1990c*; *van Groen and Wyss, 1990a*; *van Groen and Wyss, 1990b*; *van Groen and Wyss, 1992*; *Shibata and Honda, 2012*; *Vogt and Miller, 1983*) ramify in and delimit the anatomical borders of the presubiculum (*Liu et al., 2021*; *Simonnet et al., 2017*). HD signals from HD cells of the anterior thalamus project to the presubiculum (*Goodridge and Taube, 1997*) where they form synapses directly with layer 3 pyramidal cells (*Nassar et al., 2018*). The retrosplenial cortex was the most strongly labeled cortical region innervating the presubiculum. We detected retrogradely transported beads in cells of layers 2 and 5 of dysgranular RSC, and layer 5 of the granular RSC across its antero-posterior axis (*Sugar and Witter, 2016*).

We examined the anatomy and physiology of these two afferent systems to ask how visual landmark signals from the RSC are combined with HD signals relayed via the ATN. Anterograde fiber tracing, with an AAV5 construct expressing Chronos-GFP, confirmed that axons from both regions project to superficial layers of the dorsal presubiculum. No RSC projections were found in ventral PrS for injections in rostral RSC, as previously noted (*Jones and Witter, 2007*; *Kononenko and Witter, 2012*). Both RSC and ATN innervated superficial presubicular layers 1 and 3, while sparing layer 2 containing somata of calbindin-positive neurons (*Figure 1K* and *Balsamo et al., 2022*). We detected microzones containing a high density of ATN-positive fibers, but no RSC fibers, in upper layer 3. It remains to be shown how these anatomical inhomogeneities may translate to functional modularity.

## Pathway-specific functional connectivity onto PrS layer 3 neurons

Photostimulation of GFP-Chronos-expressing axons let us compare synaptic events initiated in layer 3 pyramidal cells by fibers from the ATN or the RSC in dorsal presubiculum slices (*Figure 2*). Both ATN and RSC afferents formed mono-synaptic glutamatergic connections with components mediated by NMDA and AMPA receptors. The amplitudes of synaptic currents varied between neurons, with an overall similar amplitude distribution for ATN and RSC stimulation. This may be due to variable expression levels (*Hooks et al., 2015*), or may imply variations in coupling weights across cells. ATN and RSC fiber-mediated EPSCs depressed during repetitive activation at 20 Hz. EPSPs induced firing early during repetitive stimulation. RSC inputs tended to produce more sustained EPSP dynamics and slower rise times than ATN afferents for low-intensity stimulation. ATN synapses with presubicular pyramidal cells resemble those made by thalamic afferents in the somatosensory cortex - high release

with depressing dynamics (*Gil et al., 1999*), possibly due to presynaptic expression of VGLut2 (*Liu et al., 2021*).

Dual-wavelength optogenetic stimulation was used for independent activation of intermingled afferents expressing blue-light-sensitive Chronos in ATN fibers and red-shifted Chrimson in RSC fibers (*Klapoetke et al., 2014*). Precautions were taken to avoid cross-stimulation since all channelrhodopsin variants are somewhat sensitive to blue light. With the fast, sensitive opsin Chronos expressed in ATN fibers, synaptic events were induced by very short (0.5 ms), low intensity, 0.25 mW, stimuli. For RSC fibers expressing Chrimson, using light stimuli of duration 2 ms, adjusting intensity up to 2 mW initiated synaptic events of comparable amplitude. Calibration experiments (*Figure 3—figure supplement 1*) provided strong evidence for the independence of responses. While Chrimson has slower dynamics than Chronos (*Klapoetke et al., 2014*), synaptic events induced by stimulating either opsin were similar, as we showed by swapping the two opsin variants injected in ATN and RSC fibers, respectively (*Figure 4*).

This dual opsin approach permitted independent stimulation with blue and red light pulses. Most (76%) recorded layer 3 neurons generated synaptic events in response to stimulation of both ATN and RSC fibers providing a substrate for integration of landmark information from the RSC with thalamic HD signals. Layer 3 cell firing was most effectively triggered by nearly coincident inputs (–2 to +5 ms separation, *Figure 4I and J*). This is of interest since HD representations in the AD and RSC are highly coherent (*Fallahnezhad et al., 2023*), with short intervals between spikes in these two regions (<5 ms; *van der Goes et al., 2024*). Converging inputs from thalamic and retrosplenial axons can, therefore, excite common postsynaptic presubicular layer 3 neurons with very short delays, such that coincidence detection by these cells will tend to enhance HD signals. Response dynamics to combined stimulation of both inputs at 20 Hz were maintained or facilitating, in contrast to the depressing dynamics of repetitive stimulation of one set of afferent fibers. Combined and temporally precise inputs from ATN and RSC may thus help maintain HD signaling during immobility. Our findings may well underlie the recent in vivo observation by *Siegenthaler et al., 2025* that visual objects refine HD coding, and that presubicular HD cells whose preferred firing direction corresponds to a visual landmark respond with higher firing rates.

## Nonlinear signal integration in layer 3 thalamo-recipient neurons and physiological significance for HD-to-landmark anchoring

Convergence of ATN and RSC axons onto single layer 3 pyramidal cells provides the anatomical basis of synaptic integration. Putative synaptic contacts from both afferent fiber systems were found on the basal dendrites of pyramidal cells (*Figure 3*), sometimes on the same branch. Photostimulation centered on the soma of recorded neurons predominantly activated synapses on basal dendrites. Supralinear summation could result from local spike-generating mechanisms if the activated synapses were located on a same dendritic branch (*Makarov et al., 2023*; *Poirazi and Papoutsi, 2020*; *Polsky et al., 2004*). Clustered synapses could also guide the formation of new spines and synapses during integration of landmark information in the HD signal. Learning might bind new inputs into functional synaptic clusters (*Hedrick et al., 2022*). Our small sample of layer 3 pyramidal cells suggests that both ATN and RSC axons target basal dendrites, and more RSC than ATN axons contact apical dendrites (*Figure 3*). Tests on the effects of precise, near-coincident activation of basal and apical synapses could be revealing but were not technically possible in this study. Distinct dendritic inputs to layer 3 cells may improve spatial perception (*Takahashi et al., 2016*) enhancing HD signal quality by integration with landmark information.

Our data show that EPSPs elicited by nearly coincident ATN and RSC inputs which exceed a threshold are amplified by NMDA receptor activation and voltage-gated inward dendritic currents (*Figure 5*; *Fricker et al., 2009*). From a small sample of cells, it appears that EPSP amplification may be facilitated by a reduction in synaptic inhibition (n=3; *Figure 5*), indicating that disinhibition may be permissive for supralinearity and gate firing by dynamic modulation of the balance between inhibition and excitation (*Milstein et al., 2015*). VIP-expressing interneurons, which are excited by cholinergic modulation could provide such disinhibition of the presubicular microcircuit (*Porter et al., 1999*; *Slomianka and Geneser, 1991*). The excitation-inhibition balance in pyramidal cells may become tipped towards excitation in the case of coincident, co-tuned thalamic and retrosplenial input. A 'when-to-learn' signal for HD updating in the presubiculum might function analogously to

the promotion by dopamine of associations between sensory cues and head direction cells in the fly (*Fisher et al., 2022*). It is, however, unclear whether LTP-type learning takes place in the mammalian head direction circuit. Presubicular layer 3 cells do not express the GluR1 subunit of AMPA receptors (*Martin et al., 1993*; *Ishihara and Fukuda, 2016*) that is critical for LTP expression (*Boehm et al., 2006*). The absence of GluR1 might indicate that the thalamo-presubicular synapses in layer 3 function without classical long-term synaptic plasticity. Matching directional with visual landmark information based on temporal coincidence may be sufficient. Algorithms for dynamic control of cognitive maps without synaptic plasticity (*Whittington et al., 2025* Neuron) propose that working memory is stored in neural attractor activity and updated by recurrent connections, which might generalize to the HD system.

Layer 3 pyramidal cells may be described as multi-compartment computational devices (*Häusser and Mel, 2003*; *Mel, 1993*; *Poirazi et al., 2003*; *Spruston, 2008*) which integrate HD and landmark information. ATN axons drive presubicular HD neurons (*Peyrache et al., 2015*) and RSC mixed selectivity neurons contribute visual landmark information and allocentric spatial references (*Jacob et al., 2017*; *Mitchell et al., 2018*; *Vann et al., 2009*). NMDA-mediated dendritic spikes enhance tuning selectivity in visual cortex (*Smith et al., 2013*; *Wilson et al., 2016*) and barrel cortex (*Lavzin et al., 2012*). Dendritic events in Layer 3 PrS cells may enable binding of visual landmarks with HD tuning. It is tempting to speculate that nonlinear synaptic integration and inhibitory gating may be involved in flexibly updating the allocentric direction of HD cells based on the integration of visual landmark information to the current HD signal (*Siegenthaler et al., 2025*). In the primary sensory cortex, nonlinearities act to increase perceptual threshold of sensory information (*Takahashi et al., 2016*; *Takahashi et al., 2020*). The attractor network in the PrS could thus be either stabilized or flexibly reset to external spatial cues.

Alternative models of HD-to-landmark anchoring have focused on the RSC, which also treats HD and landmark signals (*Page and Jeffery, 2018*; *Yan et al., 2021*). Possibly, the RSC registers a direction to a landmark rather than comparing it with the current HD (*Sit and Goard, 2023*). The integrated information would then reach PrS, which, in contrast to RSC, is uniquely positioned to update the signal in the LMN (*Yoder and Taube, 2011*).

What is the neuronal and circuit mechanism for integrating the 'right' set of landmarks to anchor the HD signal? In our slice work, we are blind to the exact nature of the signal that is carried by ATN and RSC axons. Critical missing information to understand the logic of HD-to-landmark anchoring also includes the degree of divergence and convergence of connections from a single thalamic or retrosplenial neuron. Do HD-tuned inputs from the thalamus converge on similarly tuned HD neurons only? Is divergence greater for the retrosplenial inputs? If so, thalamic input might pre-select a range of HD neurons, and converging RSC input might narrow down the precise HD neurons that become active (*Figure 7—figure supplement 2*). In the future, activity-dependent labeling strategies might help to tie together information on the tuning of pre-synaptic neurons and their convergence or divergence onto functionally defined postsynaptic target cells.

## Functional significance of two-layer processing preceding projection to LMN

Burst-firing layer 4 PrS cells project a distinct version of HD-landmark signals to the LMN. The dendrites of these LMN projecting neurons overlap little with direct ATN and RSC inputs granting partial isolation from direct excitation, and permitting a second level of integration of information from layer 3 neurons. Layer 3 axons may innervate layer 4 neurons basal dendrites, which may be driven to fire by sufficiently strong excitatory inputs from layer 3, especially if combined with cholinergic activation.

There are advantages to segregating the integration of converging input signals from the updating signal across layers. Segregation permits both a fast transmission of an integrated signal to the medial entorhinal cortex and the conditional transmission of an updating signal mediated by burst firing to upstream HD circuit elements. The synaptic threshold implicit in this two-stage system permits a gated updating. Functionally, layer 4 neurons are uniquely positioned to update the HD signal in the LMN with visual landmark information (*Yoder et al., 2015*). In this way, cell-type specific cholinergic facilitation may help idiothetic cue-based navigation (*Yoder et al., 2017*). These cellular and synaptic circuit data support and expand the findings of *Yoder et al., 2015*; *Yoder et al., 2017*. They clarify how the thalamic HD signal integrated with visual landmark

information is relayed to the grid cell system in the medial entorhinal cortex and to the lateral mammillary nuclei.

## Limitations

Burst firing may play a role in learning in hierarchical circuits (*Friedenberger et al., 2023*; *Payeur et al., 2021*). In the HD circuit, landmark information contained in bursts might reset or anchor the HD attractor in the LMN and beyond. The effects of burst firing signals transmitted to the LMN remain to be assessed. Potentially, HD signals may be updated with visual signals at several sites, including the PrS, but PrS uniquely provides a feedback projection to the lateral mammillary nucleus. Further work on layer 3 to layer 4 transmission is warranted, and the link to spatial perception and behavioral updating needs to be strengthened.

## Methods

**Key resources table**

| Reagent type (species) or resource | Designation | Source or reference | Identifiers | Additional information |
|---|---|---|---|---|
| Other | AAV5.Syn.Chronos-GFP.WPRE.bGH | Addgene | RRID:Addgene_59170 | 59,170P |
| Other | AAV5.Syn.ChrimsonR-tdTomato.WPRE.bGH | Addgene | RRID:Addgene_59171 | 59,171P |
| Other | pAAVretro-CAG-tdTomato | Addgene | RRID:Addgene_59462 | 59,462P |
| Chemical compound, drug | Retrobeads | Lumafluor | Red Retrobeads | |
| Peptide, recombinant protein | Streptavidin, Alexa Fluor 647 Conjugate | Thermo Fisher Scientific | S32357 | |
| Chemical compound, drug | TTX, tetrodotoxin citrate | Tocris | PubChemID:16759596 | 1069 |
| Chemical compound, drug | 4-Aminopyridine | Sigma-Aldrich | PubChemID:24891285 | A78403 |
| Chemical compound, drug | D-APV | Tocris | PubChemID:135342 | 0106 |
| Chemical compound, drug | NBQX | Abcam | PubChemID:3272523 | Ab120046 |
| Chemical compound, drug | Gabazine | Tocris | PubChemID:107895 | SR 95531 |
| Chemical compound, drug | Carbachol, carbamoylcholine chloride | Sigma | PubChemID:24277829 | C4382 |
| Strain, strain background | C57BL/6J | Janvier Labs | https://janvier-labs.com/en/fiche_produit/2_c57bl-6j_mouse/ | |
| Strain, strain background | B6.Cg-*Gt(ROSA)26Sor*^*tm14(CAG-tdTomato)Hze*/J | Jackson | RRID:IMSR_JAX:007914 | 007914 |
| Software, algorithm | MATLAB | MathWorks | RRID:SCR_001622 | |
| Software, algorithm | pClamp | Molecular devices | RRID:SCR_011323 | |
| Software, algorithm | AxoGraphX | Axograph | RRID:SCR_014284 | |
| Other | Prolong Gold antifade mountant | Invitrogen | RRID:SCR_015961 | P36930 |

## Animals

Experiments were performed on male and female wild-type and transgenic (Ai14 reporter line - Jax n007914) C57BL/6 mice of age 35–70 days, housed on a 12 hr light/dark cycle with food and water available ad libitum. Animal care and use conformed to the European Community Council Directive (2010/63/EU) and French law (87/848). Our study was approved by the local ethics committee (CEEA - 34) and the French Ministry for Research 01025.02.

## Viral vectors and beads

Projecting neurons were labeled with retrograde fluorescent tracers (Retrobeads, Lumafluor, and AAV2retro-CAG-tdTomato, Addgene 59,462P). Fluorescent beads were stored at 4°C before use. Channelrhodopsin expression was achieved by injecting adeno-associated viral constructs. AAV5. Syn.Chronos-GFP.WPRE.bGH (AAV5-Chronos, Penn Vector Core, Addgene 59,170P) was used to induce neuronal expression of the blue light-induced channelrhodopsin Chronos fused to the GFP marker and under the control of the Synapsin promoter. AAV5.Syn.ChrimsonR-tdTomato.WPRE.bGH (AAV5-Chrimson, Penn Vector Core, Addgene 59,171P) induced neuronal expression of the red light-gated channelrhodopsin Chrimson fused to the tdTomato marker, under the control of the Synapsin promoter. Viral vectors were stored at –80°C before use.

## Stereotaxic surgery

Mice at ages of 4–5 weeks were anesthetized by intraperitoneal (i.p.) injection of a mixture of ketamine hydrochloride and xylazine (100 and 15 mg/kg, respectively, in NaCl 0.9%). They were placed in a stereotaxic frame for injections. Fluorescent beads for retrograde tracing were injected (300–500 nl) into the PrS at the coordinates: –4.06 antero-posterior (AP), 2.00 medio-lateral (ML) and –2.15 mm dorso-ventral (DV), and into the LMN (–2.8 AP, 0.75 ML, –5.35 DV) with respect to the bregma. Note, due to the small size of LMN, some overflow to the medial mammillary nucleus (MMN) may have occurred.

Viral injections were performed unilaterally (*Mathon et al., 2015*; *Richevaux et al., 2019*) at the coordinates –0.82 AP, 0.75 ML, and –3.2 mm DV for the ADN, and at –2.1 to –2.15 AP, 0.65 ML, and –0.65 mm DV for the RSC. Volumes of 200–250 nl were injected with a 10 µL Hamilton syringe equipped with 33 ga needle over a time of 10 min. The needle was slowly removed after a delay of 10 min to avoid leakage from the injection site. The best expression of AAV5 serotypes was achieved after 3–4 weeks.

## Tissue fixation and slicing for retrograde tracing

Brains were removed for anatomy at 4 days after retrobead injection. Mice were anesthetized by i.p. injection of the ketamine/xylazine mixture. An intracardiac perfusion with 0.1 M PBS was followed by perfusion with 4% paraformaldehyde. Brains were stored overnight in paraformaldehyde at 4°C and then washed in PBS. Coronal or horizontal sections were cut at 100 µm with a vibratome and stored in sucrose at 4°C.

## Preparation of brain slices for physiology

Slices of the temporal lobe were prepared 3–4 weeks after injection of AAV5 viral constructions. Mice were anesthetized by i.p. injection of the ketamine/xylazine mixture. They were then perfused intracardially with a cutting solution containing (in mM): 125 NaCl, 25 sucrose, 2.5 KCl, 25 NaHCO$_3$, 1.25 NaH$_2$PO$_4$, 2.5 D-glucose, 0.1 CaCl$_2$, 7 MgCl$_2$, cooled to 4°C, and oxygenated with a 5% CO$_2$/95% O$_2$. The brain was removed and a vibratome was used to cut horizontal slices at 300 µm in the same solution. Slices were stored for 15 min at 34°C in an ACSF containing (in mM): 124 NaCl, 2.5 KCl, 26 NaHCO$_3$, 1 NaH$_2$PO$_4$, 2 CaCl$_2$, 2 MgCl$_2$, and 11 D-glucose, bubbled with 5% CO$_2$/95% O$_2$. They were then kept in the same solution at room temperature until recording.

## Whole-cell patch-clamp recordings

Slices were transferred to a recording chamber perfused with oxygenated, warmed (~32°C) ACSF mounted on an epifluorescence microscope. Patch-clamp records were made from neurons with borosilicate glass pipettes of external diameter 1.5 mm (Clark Capillary Glass, Harvard Apparatus) pulled with a Brown-Flaming electrode puller (Sutter Instruments). Electrodes, filled with a potassium-gluconate-based solution containing (in mM): 135 K-gluconate, 1.2 KCl, 10 HEPES, 0.2 EGTA, 2 MgCl$_2$, 4 MgATP, 0.4 Tris-GTP, and 10 Na$_2$-phosphocreatine, had a resistance of 4–8 MΩ. An alternative, cesium-gluconate-based solution facilitated neuronal depolarization to examine synaptic inhibition, in *Figure 5*. It contained (in mM): 125 Cs-gluconate, 10 HEPES, 0.2 EGTA, 2 MgCl$_2$, 4 MgATP, 0.4 Tris-GTP, and 10 Na$_2$-Phosphocreatine, together with 5 mM QX-314 to block Na$^+$ channels. Pipette solutions also contained 3 mM biocytin to reveal morphology after recording. They were adjusted to pH 7.3 and osmolarity 290 mOsm. Whole-cell, patch-clamp signals were filtered at 3 kHz, amplified

with a MultiClamp 700B amplifier and acquired with pCLAMP software (Molecular Devices). In a subset of experiments, the following drugs were used to modulate the responses to optogenetic stimulations; the presence of these drugs is indicated in the figure and figure legend, whenever applicable. Monosynaptic excitation induced by optical stimulation was tested in the presence of TTX (1 μM) and 4-AP (100 μM). NBQX (10 μM) and APV (100 μM) were used to block AMPA and NMDA receptors, respectively. Gabazine (10 μM) was used to block GABA$_A$ receptors. Carbachol (10 μM) was used to activate acetylcholine receptors. All drugs were bath applied.

## Optical stimulation

LED illumination (Cairn Research, OptoLED) was used to visualize the fluorescent reporters GFP and tdTomato, and to stimulate opsin-expressing axons, using a 470 nm LED for Chronos and a 627 nm LED for Chrimson. Illuminated spots had a diameter of 200 μm with a 60x objective and were centered on the recorded cell soma. Photostimulation thus covered most of the basilar dendrites of layer 3 pyramidal neurons (typical distance from tip of apical to tip of basilar dendrite <400 μm). Stimuli consisted of light pulses of 0.5–5 ms duration, repeated 5–10 times at 20 Hz.

A multiband filter allowed simultaneous stimulation by blue and red LEDs of axons containing Chronos and Chrimson (*Simonnet et al., 2021*). Stimulus power intensity (set in mV) was calibrated as light intensity. The response probability of layer 3 cells was calibrated to blue or red illumination of ATN or RSC afferents expressing Chronos or Chrimson to avoid stimulus overlap (*Figure 3—figure supplement 1*). Chronos was targeted to the ATN and Chrimson to the RSC after testing the reverse configuration. Projections from the thalamus are larger and Chronos is more sensitive to blue light, so this configuration assured reliable activation of thalamic fibers at minimal blue-light intensities. For experiments investigating the integration of ATN and RSC inputs, we aimed to give similar weights to both inputs: Chrimson-expressing fibers were stimulated with light of intensity adjusted to initiate optically evoked excitatory postsynaptic potentials (oEPSPs) of amplitude similar to those induced by blue light Chronos-fiber stimuli.

## Data analysis

ATN and RSC projections to the PrS were analyzed from Chronos and Chrimson expression, using the ImageJ Plot Profile plug-in to quantitate normalized plot profiles (2000 pixels) of horizontal presubicular sections. Dorso-ventral differences were derived by dividing differences in labeling intensity between the PrS and the dentate gyrus (DG) molecular layer by DG intensity in slices from five dorso-ventral PrS levels. Values from all animals were averaged and then normalized.

Cells were qualified as synaptically 'connected' when they responded to light stimulation of afferent axons with a delay <8 ms. Slices with very low or absent expression of the fluorescent reporter were excluded. Cells were qualified as 'non-connected' synaptically if they did not respond to light stimulation, but at least one neighboring cell in the same slice did.

Intrinsic neuronal properties were analyzed with custom MATLAB routines to derive 15 electrophysiological parameters (*Huang et al., 2017*). Parameters were standardized and unsupervised cluster analysis performed with MATLAB was used to compare different neurons (*Huang et al., 2017*; *Simonnet et al., 2013*).

Axograph was used to analyze responses to optical stimulation of ATN and RSC fibers. Layer 3 neurons were recorded at potentials near –65 mV. Responses to light were averaged from 10 stimulus trains at 20 Hz. Amplitudes and latencies of initial light-evoked EPSCs, of latency shorter than 10 ms, were quantified from voltage-clamp records. Latency was measured from stimulus onset to 10% of the peak amplitude of the first optically induced EPSC. Amplitude was measured with respect to the pre-stimulus baseline. Paired-pulse ratio (PPR) was defined as the ratio of the amplitude of the second to the first EPSC and 10/1 ratio as that between the 10[th] and the first EPSC, in responses to 20 Hz stimulations. Spike probability was counted over 5–10 trials per cell and then averaged over all cells. EPSPs induced by dual-wavelength stimulation were analyzed using Axograph and a custom-made software. Events evoked by light stimuli were detected in a window of 1–10 ms after stimulation. EPSP amplitude and integrated area were calculated over 50 ms after stimulation. Baseline suppression was applied using an average of membrane potential during 50 ms before stimulation. The summation of ATN and RSC-evoked EPSCs in layer 3 neurons was determined from the amplitude and integral of

averaged events. Summation was considered to be supralinear if values were more than 10% higher than a linear addition, linear for values within ±10%, and sublinear for values more than 10% lower.

## Biocytin revelation and morphology

Recorded neurons were filled with biocytin to visualize their morphology and location. Slices were fixed in 4% PFA in 0.1 M PBS at 4°C overnight, washed three times in 0.1 PBS and cryoprotected in 30% sucrose. Cell membranes were permeabilized by three freeze-thaw cycles over dry ice and rinsed in PBS. Slices were agitated in a saturation buffer (2% milk, 1% Triton X-100 in 0.1 M PBS) for 2 hr at room temperature. They were then incubated with Streptavidin-Cy5 conjugate (1:500) and DAPI (1:1000) overnight at 4°C. Sections were washed 3 times with PBS and mounted on coverslips with ProLong Gold antifade mountant. For anatomical study, they were mounted in Mowiol medium. Cell, bead, and virus expression were visualized with an epifluorescence microscope. Higher resolution of morphology was obtained with confocal microscopy. Cell morphologies were reconstructed using IMARIS.

## Quantification and statistical analysis

Data was analyzed with AxoGraphX and custom-written software (MATLAB, The MathWorks). Results are given as mean ± SEM. Statistical analysis was performed with Prism (GraphPad Software). The Mann-Whitney unpaired t-test was used to compare two groups. The Wilcoxon or Kruskal-Wallis test was used to compare paired groups. Evolution of parameters evoked by optical stimulation (current or potential amplitudes, spike probability) was analyzed with Friedman's test followed by multiple comparison Dunn's test. Šidák's multiple comparison test was also used to compare linear and observed responses to ATN and RSC stimulations. Significance levels are given as p-values.

## Acknowledgements

This work is supported by the Centre National de la Recherche Scientifique and the Université Paris Cité. DF received funding from the Agence Nationale de la Recherche (ANR-DFG Program, ANR-18-CE92-0051 BURST), from the ERA-NET NEURON Program (ANR-20-NEUR-0005 VELOSO), and the FLAG-ERA HBP Program (ANR-21-HBPR-0002 VIPattract). This work has benefited from support by the BioMedTech Facilities at Université Paris Cité (Institut National de la Santé et de la Recherche Médicale Unité S36/Unité Mixte de Service 2009). We thank Dr. Li-Wen Huang for help with analysis routines. We thank Dr. Kate Jeffery and the Royal Society COST-share program, technical assistance from Fabrice Licata and Dr. Abdelali Jalil from the UPC Microscopy platform, Claire Lovo from ICM Quant, Dr. Boris Lamotte d'Incamps, and Dr. Richard Miles for helpful comments on the manuscript.

## Additional information

### Funding

| Funder | Grant reference number | Author |
| --- | --- | --- |
| Agence Nationale de la Recherche | ANR-18-CE92-0051 | Desdemona Fricker |
| ERA-Net NEURON | ANR-20-NEUR-0005 | Desdemona Fricker |
| FLAG-ERA HBP | ANR-21-HBPR-0002 | Desdemona Fricker |
| Ministère de l'Enseignement Supérieur et de la Recherche | PhD fellowship | Louis Richevaux |

The funders had no role in study design, data collection and interpretation, or the decision to submit the work for publication.

### Author contributions

Louis Richevaux, Conceptualization, Data curation, Formal analysis, Investigation, Visualization, Methodology, Writing – original draft, Writing – review and editing; Dongkyun Lim, Léa Dias Rodrigues,

Constanze Mauthe, Nathalie Sol-Foulon, Investigation, Methodology; Mérie Nassar, Formal analysis, Investigation, Methodology; Ivan Cohen, Software, Methodology; Desdemona Fricker, Conceptualization, Supervision, Funding acquisition, Validation, Investigation, Methodology, Writing – original draft, Project administration, Writing – review and editing

**Author ORCIDs**
Louis Richevaux ![orcid] https://orcid.org/0000-0001-7837-3489
Dongkyun Lim ![orcid] https://orcid.org/0009-0004-8756-6505
Desdemona Fricker ![orcid] https://orcid.org/0000-0001-7328-9480

**Ethics**
Animal care and use conformed to the European Community Council Directive (2010/63/EU) and French law (87/848). Our study was approved by the local ethics committee (CEEA - 34) and the French Ministry for Research 01025.02.

Reviewer #1 (Public review): https://doi.org/10.7554/eLife.92443.3.sa1
Reviewer #2 (Public review): https://doi.org/10.7554/eLife.92443.3.sa2
Reviewer #3 (Public review): https://doi.org/10.7554/eLife.92443.3.sa3
Author response https://doi.org/10.7554/eLife.92443.3.sa4

## Additional files

### Supplementary files
MDAR checklist

### Data availability
Data are stored on local servers at the Université Paris Cité and available on https://doi.org/10.5061/dryad.9ghx3ffxd. The Matlab code for analysis of intrinsic properties is available at https://github.com/schoki0710/Intrinsic_Properties (copy archived at *schoki0710, 2025*).

The following dataset was generated:

| Author(s) | Year | Dataset title | Dataset URL | Database and Identifier |
|---|---|---|---|---|
| Richevaux L, Lim D, Nassar M, Dias Rodrigues L, Mauthe C, Cohen I, Sol-Foulon N, Fricker D | 2025 | Projection-specific integration of convergent thalamic and retrosplenial signals in the presubicular head direction cortex | https://doi.org/10.5061/dryad.9ghx3ffxd | Dryad Digital Repository, 10.5061/dryad.9ghx3ffxd |

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
