## [Editor Report · eLife Assessment]

This **valuable** study combines anatomical tracing and slice physiology to examine how anterior thalamic and retrosplenial inputs converge in the presubiculum, a key region in the navigation circuit. The authors show that near-simultaneous co-activation of retrosplenial and thalamic inputs drives supra-linear presubiculum responses, revealing a potential cellular mechanism for anchoring the brain's head direction system to external visual landmarks. Their thorough experimental approach and analyses provide **convincing** evidence for the cellular basis of how the brain's internal compass may be anchored to the external world, laying the groundwork for future experimental testing in vivo.

---

## [Referee Report · Reviewer #1 (Public review)]

Summary:

In this manuscript, the authors use anatomical tracing and slice physiology to investigate the integration of thalamic (ATN) and retrosplenial cortical (RSC) signals in the dorsal presubiculum (PrS). This work will be of interest to the field, as postsubiculum is thought to be a key region for integrating internal head direction representations with external landmarks. The main result is that ATN and RSC inputs drive the same L3 PrS neurons, which exhibit superlinear summation to near-coincident inputs. Moreover, this activity can induce bursting in L4 PrS neurons, which can pass the signals LMN (perhaps gated by cholinergic input).

Strengths:

The slice physiology experiments are carefully done. The analyses are clear and convincing, and the figures and results are well composed. Overall, these results will be a welcome addition to the field.

Weaknesses:

The conclusions about the circuit-level function of L3 PrS neurons sometimes outstrip the data, and their model of the integration of these inputs is unclear. I would recommend some revision of the introduction and discussion. I also had some minor comments about the experimental details and analysis.

Specific major comments:

(1) I found that the authors' claims sometimes outstrip their data, given that there were no in vivo recordings during behavior. For example, in the abstract that their results indicate "that layer 3 neurons can transmit a visually matched HD signal to medial entorhinal cortex", and in the conclusion they state "[...] cortical RSC projections that carry visual landmark information converge on layer 3 pyramidal cells of the dorsal presubiculum". However, they never measured the nature of the signals coming from ATN and RSC to L3 PrS (or signals sent to downstream regions). Their claim is somewhat reasonable with respect to ATN, where the majority of neurons encode HD, but neurons in RSC encode a vast array of spatial and non-spatial variables other than landmark information (e.g., head direction, egocentric boundaries, allocentric position, spatial context, task history to name a few), so making strong claims about the nature of the incoming signals is unwarranted.

(2) Related to the first point, the authors hint at, but never explain, how coincident firing of ATN and RSC inputs would help anchor HD signals to visual landmarks. Although the lesion data (Yoder et al. 2011 and 2015) support their claims, it would be helpful if the proposed circuit mechanism was stated explicitly (a schematic of their model would be helpful in understanding the logic). For example, how do neurons integrate the "right" sets of landmarks and HD signals to ensure a stable anchoring? Moreover, it would be helpful to discuss alternative models of HD-to-landmark anchoring, including several studies that have proposed that the integration may (also?) occur in RSC (Page & Jeffrey, 2018; Yan, Burgess, Bicanski, 2021; Sit & Goard, 2023). Currently, much of the Discussion simply summarizes the results of the study, this space could be better used in mapping the findings to the existing literature on the overarching question of how HD signals are anchored to landmarks.

Comments on revised version:

The authors addressed all of my major points and most of my minor points in the revised submission.

---

## [Referee Report · Reviewer #2 (Public review)]

Richevaux et al investigate how anterior thalamic (AD) and retrosplenial (RSC) inputs are integrated by single presubicular (PrS) layer 3 neurons. They show that these two inputs converge onto single PrS layer 3 principal cells. By performing dual wavelength photostimulation of these two inputs in horizontal slices, the authors show that in most layer 3 cells, these inputs summate supra-linearly. They extend the experiments by focusing on putative layer 4 PrS neurons and show that they do not receive direct anterior thalamic nor retrosplenial inputs; rather, they are (indirectly) driven to burst firing in response to strong activation of the PrS network.

This is a valuable study, which investigates an important question - how visual landmark information (possibly mediated by retrosplenial inputs) converges and integrates with HD information (conveyed by the AD nucleus of the thalamus) within PrS circuitry. The data indicate that near-coincident activation of retrosplenial and thalamic inputs leads to non-linear integration in target layer 3 neurons, thereby offering a potential biological basis for landmark + HD binding.

Main limitations relate to the anatomical annotation of 'putative' PrS L4 neurons, and to the presentation of retrosplenial / thalamic input modularity. Specifically, more evidence should be provided to convincingly demonstrate that the 'putative L4 neurons' of the PrS are not distal subicular neurons (as the authors' anatomy and physiology experiments seem to indicate). The modularity of thalamic and retrosplenial inputs could be better clarified in relation to the known PrS modularity.

---

## [Referee Report · Reviewer #3 (Public review)]

Summary:

The authors sought to determine, at the level of individual presubiculum pyramidal cells, how allocentric spatial information from retrosplenial cortex was integrated with egocentric information from the anterior thalamic nuclei. Employing a dual opsin optogenetic approach with patch clamp electrophysiology, Richevaux and colleagues found that around three quarters of layer 3 pyramidal cells in presubiculum receive monosynaptic input from both brain regions. While some interesting questions remain e.g. the role of inhibitory interneurons in gating the information flow and through different layers of presubiculum, this paper provides valuable insights into the microcircuitry of this brain region and the role that it may play in spatial navigation.

Strengths:

One of the main strengths of this manuscript was that the dual opsin approach allowed the direct comparison of different inputs within an individual neuron, helping to control for what might otherwise have been an important source of variation. The experiments were well-executed and the data rigorously analysed. The conclusions were appropriate to the experimental questions and were well-supported by the results. These data will help to inform in vivo experiments aimed at understanding the contribution of different brain regions in spatial navigation and could be valuable for computational modelling.

Weaknesses:

Some attempts were made to gain mechanistic insights into how inhibitory neurotransmission may affect processing in presubiuclum (e.g. figure 5) but these experiments were a little underpowered and the analysis carried out could have been more comprehensively undertaken, as was done for other experiments in the manuscript.

Comments on revised version:

The authors have addressed all of my comments and I have nothing further to add. Well done for an interesting and valuable contribution!

---

## [Author Response]

The following is the authors’ response to the original reviews.

**Reviewer #1 (Public Review):**
Summary:In this manuscript, the authors use anatomical tracing and slice physiology to investigate the integration of thalamic (ATN) and retrosplenial cortical (RSC) signals in the dorsal presubiculum (PrS). This work will be of interest to the field, as the postsubiculum is thought to be a key region for integrating internal head direction representations with external landmarks. The main result is that ATN and RSC inputs drive the same L3 PrS neurons, which exhibit superlinear summation to near-coincident inputs. Moreover, this activity can induce bursting in L4 PrS neurons, which can pass the signals LMN (perhaps gated by cholinergic input).Strengths:The slice physiology experiments are carefully done. The analyses are clear and convincing, and the figures and results are well-composed. Overall, these results will be a welcome addition to the field.

We thank this reviewer for the positive comment on our work.

Weaknesses:The conclusions about the circuit-level function of L3 PrS neurons sometimes outstrip the data, and their model of the integration of these inputs is unclear. I would recommend some revision of the introduction and discussion. I also had some minor comments about the experimental details and analysis.Specific major comments:(1) I found that the authors' claims sometimes outstrip their data, given that there were no in vivo recordings during behavior. For example, in the abstract, their results indicate "that layer 3 neurons can transmit a visually matched HD signal to medial entorhinal cortex", and in the conclusion they state "[...] cortical RSC projections that carry visual landmark information converge on layer 3 pyramidal cells of the dorsal presubiculum". However, they never measured the nature of the signals coming from ATN and RSC to L3 PrS (or signals sent to downstream regions). Their claim is somewhat reasonable with respect to ATN, where the majority of neurons encode HD, but neurons in RSC encode a vast array of spatial and non-spatial variables other than landmark information (e.g., head direction, egocentric boundaries, allocentric position, spatial context, task history to name a few), so making strong claims about the nature of the incoming signals is unwarranted.

We agree of course that RSC does not only encode landmark information. We have clarified this point in the introduction (line 69-70) and formulated more carefully in the abstract (removed the word ‘landmark’ in line 17) and in the introduction (line 82-83). In the discussion we explicitly state that ‘In our slice work we are blind to the exact nature of the signal that is carried by ATN and RSC axons’ (line 522-523).

(2) Related to the first point, the authors hint at, but never explain, how coincident firing of ATN and RSC inputs would help anchor HD signals to visual landmarks. Although the lesion data (Yoder et al. 2011 and 2015) support their claims, it would be helpful if the proposed circuit mechanism was stated explicitly (a schematic of their model would be helpful in understanding the logic). For example, how do neurons integrate the "right" sets of landmarks and HD signals to ensure stable anchoring? Moreover, it would be helpful to discuss alternative models of HD-to-landmark anchoring, including several studies that have proposed that the integration may (also?) occur in RSC (Page & Jeffrey, 2018; Yan, Burgess, Bicanski, 2021; Sit & Goard, 2023). Currently, much of the Discussion simply summarizes the results of the study, this space could be better used in mapping the findings to the existing literature on the overarching question of how HD signals are anchored to landmarks.

We agree with the reviewer on the importance of the question, how do neurons integrate the “right” sets of landmarks and HD signals to ensure stable anchoring? Based on our results we provide a schematic to illustrate possible scenarios, and we include it as a supplementary figure (Figure 1, to be included in the ms as Figure 7—figure supplement 2), as well as a new paragraph in the discussion section (line 516-531). We point out that critical information on the convergence and divergence of functionally defined inputs is still lacking, both for principal cells and interneurons

Interestingly, recent evidence from functional ultrasound imaging and electrical single cell recording demonstrated that visual objects may refine head direction coding, specifically in the dorsal presubiculum (Siegenthaler et al. bioRxiv 2024.10.21.619417; doi: https://doi.org/10.1101/2024.10.21.619417). The increase in firing rate for HD cells whose preferred firing direction corresponds to a visual landmark could be supported by the supralinear summation of thalamic HD signals and retrosplenial input described in our study. We include this point in the discussion (line 460-462), and hope that our work will spur further investigations.

**Reviewer #2 (Public Review):**
Richevaux et al investigate how anterior thalamic (AD) and retrosplenial (RSC) inputs are integrated by single presubicular (PrS) layer 3 neurons. They show that these two inputs converge onto single PrS layer 3 principal cells. By performing dual-wavelength photostimulation of these two inputs in horizontal slices, the authors show that in most layer 3 cells, these inputs summate supra-linearly. They extend the experiments by focusing on putative layer 4 PrS neurons, and show that they do not receive direct anterior thalamic nor retrosplenial inputs; rather, they are (indirectly) driven to burst firing in response to strong activation of the PrS network.This is a valuable study, that investigates an important question - how visual landmark information (possibly mediated by retrosplenial inputs) converges and integrates with HD information (conveyed by the AD nucleus of the thalamus) within PrS circuitry. The data indicate that near-coincident activation of retrosplenial and thalamic inputs leads to non-linear integration in target layer 3 neurons, thereby offering a potential biological basis for landmark + HD binding.The main limitations relate to the anatomical annotation of 'putative' PrS L4 neurons, and to the presentation of retrosplenial/thalamic input modularity. Specifically, more evidence should be provided to convincingly demonstrate that the 'putative L4 neurons' of the PrS are not distal subicular neurons (as the authors' anatomy and physiology experiments seem to indicate). The modularity of thalamic and retrosplenial inputs could be better clarified in relation to the known PrS modularity.

We thank the reviewer for their important feedback. We discuss what defines presubicular layer 4 in horizontal slices, cite relevant literature, and provide new and higher resolution images. See below for detailed responses to the reviewer’s comments, in the section ‘recommendations to authors’.

**Reviewer #3 (Public Review):**
Summary:The authors sought to determine, at the level of individual presubiculum pyramidal cells, how allocentric spatial information from the retrosplenial cortex was integrated with egocentric information from the anterior thalamic nuclei. Employing a dual opsin optogenetic approach with patch clamp electrophysiology, Richevaux, and colleagues found that around three-quarters of layer 3 pyramidal cells in the presubiculum receive monosynaptic input from both brain regions. While some interesting questions remain (e.g. the role of inhibitory interneurons in gating the information flow and through different layers of presubiculum, this paper provides valuable insights into the microcircuitry of this brain region and the role that it may play in spatial navigation).Strengths:One of the main strengths of this manuscript was that the dual opsin approach allowed the direct comparison of different inputs within an individual neuron, helping to control for what might otherwise have been an important source of variation. The experiments were well-executed and the data was rigorously analysed. The conclusions were appropriate to the experimental questions and were well-supported by the results. These data will help to inform in vivo experiments aimed at understanding the contribution of different brain regions in spatial navigation and could be valuable for computational modelling.Weaknesses:Some attempts were made to gain mechanistic insights into how inhibitory neurotransmission may affect processing in the presubiculum (e.g. Figure 5) but these experiments were a little underpowered and the analysis carried out could have been more comprehensively undertaken, as was done for other experiments in the manuscript.

We agree that the role of interneurons for landmark anchoring through convergence in Presubiculum requires further investigation. In our latest work on the recruitment of VIP interneurons we begin to address this point in slices (Nassar et al., 2024 Neuroscience. doi: 10.1016/j.neuroscience.2024.09.032.); more work in behaving animals will be needed.

**Reviewer #1 (Recommendations For The Authors):**
Full comments below. Beyond the (mostly minor) issues noted below, this is a very well-written paper and I look forward to seeing it in print.Major comments:(1) I found that the authors' claims sometimes outstrip their data, given that there were no in vivo recordings during behavior. For example, in the abstract, their results indicate "that layer 3 neurons can transmit a visually matched HD signal to medial entorhinal cortex", and in the conclusion they state "[...] cortical RSC projections that carry visual landmark information converge on layer 3 pyramidal cells of the dorsal presubiculum". However, they never measured the nature of the signals coming from ATN and RSC to L3 PrS (or signals sent to downstream regions). Their claim is somewhat reasonable with respect to ATN, where the majority of neurons encode HD, but neurons in RSC encode a vast array of spatial and non-spatial variables other than landmark information (e.g., head direction, egocentric boundaries, allocentric position, spatial context, task history to name a few), so making strong claims about the nature of the incoming signals is unwarranted.

Our study was motivated by the seminal work from Yoder et al., 2011 and 2015, indicating that visual landmark information is processed in PoS and from there transmitted to the LMN. Based on that, and in the interest of readability, we may have used an oversimplified shorthand for the type of signal carried by RSC axons. There are numerous studies indicating a role for RSC in encoding visual landmark information (Auger et al., 2012; Jacob et al., 2017; Lozano et al., 2017; Fischer et al., 2020; Keshavarzi et al., 2022; Sit and Goard, 2023); we agree of course that this is certainly not the only variable that is represented. Therefore we change the text to make this point clear:

Abstract, line 17: removed the word ‘landmark’

Introduction, line 69: added “...and supports an array of cognitive functions including memory, spatial and non-spatial context and navigation (Vann et al., 2009; Vedder et al., 2017). ”

Introduction, line 82: changed “...designed to examine the convergence of visual landmark information, that is possibly integrated in the RSC, and vestibular based thalamic head direction signals”.

Discussion, line 522-523: added “In our slice work we are blind to the exact nature of the signal that is carried by ATN and RSC axons.”

(2) Related to the first point, the authors hint at, but never explain, how coincident firing of ATN and RSC inputs would help anchor HD signals to visual landmarks. Although the lesion data (Yoder et al., 2011 and 2015) support their claims, it would be helpful if the proposed circuit mechanism was stated explicitly (a schematic of their model would be helpful in understanding the logic). For example, how do neurons integrate the "right" sets of landmarks and HD signals to ensure stable anchoring? Moreover, it would be helpful to discuss alternative models of HD-to-landmark anchoring, including several studies that have proposed that the integration may (also?) occur in RSC (Page & Jeffrey, 2018; Yan, Burgess, Bicanski, 2021; Sit & Goard, 2023). Currently, much of the Discussion simply summarizes the results of the study, this space could be better used in mapping the findings to the existing literature on the overarching question of how HD signals are anchored to landmarks.

We suggest a physiological mechanism for inputs to be selectively integrated and amplified, based on temporal coincidence. Of course there are still many unknowns, including the divergence of connections from a single thalamic or retrosplenial input neuron. The anatomical connectivity of inputs will be critical, as well as the subcellular arrangement of synaptic contacts. Neuromodulation and changes in the balance of excitation and inhibition will need to be factored in. While it is premature to provide a comprehensive explanation for landmark anchoring of HD signals in PrS, our results have led us to include a schematic, to illustrate our thinking (Figure 1, see below).

Do HD tuned inputs from thalamus converge on similarly tuned HD neurons only? Is divergence greater for the retrosplenial inputs? If so, thalamic input might pre-select a range of HD neurons, and converging RSC input might narrow down the precise HD neurons that become active (Figure 1). In the future, the use of activity dependent labeling strategies might help to tie together information on the tuning of pre-synaptic neurons, and their convergence or divergence onto functionally defined postsynaptic target cells. This critical information is still lacking, for principal cells, and also for interneurons.

Interneurons may have a key role in HD-to-landmark anchoring. SST interneurons support stability of HD signals (Simonnet et al., 2017) and VIP interneurons flexibly disinhibit the system (Nassar et al., 2024). Could disinhibition be a necessary condition to create a window of opportunity for updating the landmark anchoring of the attractor? Single PV interneurons might receive thalamic and retrosplenial inputs non-specifically. We need to distinguish the conditions for when the excitation-inhibition balance in pyramidal cells may become tipped towards excitation, and the case of coincident, co-tuned thalamic and retrosplenial input may be such a condition. Elucidating the principles of hardwiring of inputs, as for example, selective convergence, will be necessary. Moreover, neuromodulation and oscillations may be critical for temporal coordination and precise temporal matching of HD-to-landmark signals.

We note that matching directional with visual landmark information based on temporal coincidence as described here does not require synaptic plasticity. Algorithms for dynamic control of cognitive maps without synaptic plasticity have been proposed (Whittington et al., 2025, Neuron): information may be stored in neural attractor activity, and the idea that working memory may rely on recurrent updates of neural activity might generalize to the HD system. We include these considerations in the discussion (line 497-501; 521-531) and hope that our work will spur further experimental investigations and modeling work.

While the focus of our work has been on PrS, we agree that RSC also treats HD and landmark signals. Possibly the RSC registers a direction to a landmark rather than comparing it with the current HD (Sit & Goard, 2023). We suggest that this integrated information then reaches PrS. In contrast to RSC, PrS is uniquely positioned to update the signal in the LMN (Yoder et al., 2011), cf. discussion (line 516-520).

Minor comments:(1) Fig 1 - Supp 1: It appears there is a lot of input to PrS from higher visual regions, could this be a source of landmark signals?

Yes, higher visual regions projecting to PrS may also be a source of landmark information, even if the visual signal is not integrated with HD at that stage (Sit & Goard 2023). The anatomical projection from the visual cortex was first described by Vogt & Miller (1983), but not studied on a functional level so far.

(2) Fig 2F, G: Although the ATN and RSC measurements look quite similar, there are no stats included. The authors should use an explicit hypothesis test.

We now compare the distributions of amplitudes and of latencies, using the Mann-Whitney U test. No significant difference between the two groups were found. Added in the figure legend: 2F, “Mann-Whitney U test revealed no significant difference (p = 0.95)”. 2G, “Mann-Whitney U test revealed no significant difference (p = 0.13)”.

(3) Fig 2 - Supp 2A, C: Again, no statistical tests. This is particularly important for panel A, where the authors state that the latencies are similar but the populations appear to be different.

Inputs from ATN and RSC have a similar ‘jitter’ (latency standard deviation) and ‘tau decay’. We added in the Fig 2 - Supp 2 figure legend: A, “Mann-Whitney U test revealed no significant difference (p = 0.26)”. C, “Mann-Whitney U test revealed no significant difference (p = 0.87)”.

As a complementary measure for the reviewer, we performed the Kolmogorov-Smirnov test which confirmed that the populations’ distributions for ‘jitter’ were not significantly different, p = 0.1533.

(4) Fig 4E, F: The statistics reporting is confusing, why are asterisks above the plots and hashmarks to the side?

Asterisks refer to a comparison between ‘dual’ and ‘sum’ for each of the 5 stimulations in a Sidak multiple comparison test. Hashmarks refer to comparison of the nth stimulation to the 1st one within dual stimulation events (Friedman + Dunn’s multiple comparison test). We mention the two-way ANOVA p-value in the legend (Sum v Dual, for both Amplitude and Surface).

(5) Fig 5C: I was confused by the 2*RSC manipulation. How do we know if there is amplification unless we know what the 2*RSC stim alone looks like?

We now label the right panel in Fig 5C as “high light intensity” or “HLI”. Increasing the activation of Chrimson increases the amplitude of the summed EPSP that now exceeds the threshold for amplification of synaptic events. Amplification refers to the shape of the plateau-like prolongation of the peak, most pronounced on the second EPSP, now indicated with an arrow. We clarify this also in the text (line 309-310).

(6) Fig 6D (supplement 1): Typo, "though" should be "through"

Yes, corrected (line 1015).

(7) Fig 6G (supplement 1): Typo, I believe this refers to the dotted are in panel F, not panel A.

Yes, corrected (line 1021).

(8) Fig 7: The effect of muscarine was qualitatively described in the Results, but there is no quantification and it is not shown in the Figure. The results should either be reported properly or removed from the Results.

We remove the last sentence in the Results.

(9) Methods: The age and sex of the mice should be reported. Transgenic mouse line should be reported (along with stock number if applicable).

We used C57BL6 mice with transgenic background (Ai14 mice, Jax n007914 reporter line) or C57BL6 wild type mice. This is now indicated in the Methods (lines 566-567).

(10) Methods: If the viruses are only referred to with their plasmid number, then the capsid used for the viruses should be specified. For example, I believe the AAV-CAG-tomato virus used the retroAAV capsid, which is important to the experiment.

Thank you for pointing this out. Indeed the AAV-CAG-tdTom virus used the retroAAV capsid, (line 575).

(11) Data/code availability: I didn't see any sort of data/code availability statement, will the data and code be made publicly available?

Data are stored on local servers at the SPPIN, Université Paris Cité, and are made available upon reasonable request. Code for intrinsic properties analysis is available on github (https://github.com/schoki0710/Intrinsic_Properties). This information is now included (line 717-720).

(12) Very minor (and these might be a matter of opinion), but I believe "records" should be "recordings", and "viral constructions" should be "viral constructs".

The text had benefited from proofreading by Richard Miles, who always preferred “records” to “recordings” in his writings. We choose to keep the current wording.

**Reviewer #2 (Recommendations For The Authors):**
Below are two major points that require clarification.(1) In the last set of experiments presented by the authors (Figs 6 onwards) they focus on 'putative L4' PrS cells. For several lines of evidence (outlined below), I am convinced that these neurons are not presubicular, but belong to the subiculum. I think this is a major point that requires substantial clarification, in order to avoid confusion in the field (see also suggestions on how to address this comment at the end of this section).Several lines of evidence support the interpretation that, what the authors call 'L4 PrS neurons', are distal subicular cells:(1.1) The anatomical location of the retrogradely-labelled cells (from mammillary bodies injections), as shown in Figs 6B, C, and Fig. 6_1B, very clearly indicates that they belong to the distal subiculum. The subicular-to-PrS boundary is a sharp anatomical boundary that follows exactly the curvature highlighted by the authors' red stainings. The authors could also use specific subicular/PrS markers to visualize this border more clearly - e.g. calbindin, Wfs-1, Zinc (though I believe this is not strictly necessary, since from the pattern of AD fibers, one can already draw very clear conclusions, see point 1.3 below).

Our criteria to delimit the presubiculum are the following: First and foremost, we rely on the defining presence of antero-dorsal thalamic fibers that target specifically the presubiculum and not the neighbouring subiculum (Simonnet et al., 2017, Nassar et al., 2018, Simonnet and Fricker, 2018; Jiayan Liu et al., 2021). This provides the precise outline of the presubicular superficial layers 1 to 3. It may have been confusing to the reviewer that our slicing angle gives horizontal sections. In fact, horizontal sections are favourable to identify the layer structure of the PrS, based on DAPI staining and the variations in cell body size. The work by Ishihara and Fukuda (2016) illustrates in their Figure 12 that the presubicular layer 4 lies below the presubicular layer 3, and forms a continuation with the subiculum (Sub1). Their Figure 4 indicates with a dotted line the “generally accepted border between the (distal) subiculum and PreS”, and it runs from the proximal tip of superficial cells of the PrS toward the white matter, among the radial direction of the cortical tissue. We agree with this definition. Others have sliced coronally (Cembrowski et al., 2018) which renders a different visualization of the border region with the subiculum.

Second, let me explain the procedure for positioning the patch electrode in electrophysiological experiments on horizontal presubicular slices. Louis Richevaux, the first author, who carried out the layer 4 cell recordings, took great care to stay very close (<50 µm) to the lower limit of the zone where the GFP labeled thalamic axons can be seen. He was extremely meticulous about the visualization under the microscope, using LED illumination, for targeting. The electrophysiological signature of layer 4 neurons with initial bursts (but not repeated bursting, in mice) is another criterion to confirm their identity (Huang et al., 2017). Post-hoc morphological revelation showed their apical dendrites, running toward the pia, sometimes crossing through the layer 3, sometimes going around the proximal tip, avoiding the thalamic axons (Figure 6D). For example the cell in Figure 6, suppl. 1 panel D, has an apical dendrite that runs through layer 3 and layer 1.

Third, retrograde labeling following stereotaxic injection into the LMN is another criterion to define PrS layer 4. This approach is helpful for visualization, and is based on the defining axonal projection of layer 4 neurons (Yoder and Taube, 2011; Huang et al., 2017). Due to the technical challenge to stereotaxically inject only into LMN, the resultant labeling may not be limited to PrS layer 4. We cannot entirely exclude some overflow of retrograde tracers (B) or retrograde virus (C) to the neighboring MMN. This would then lead to co-labeling of the subiculum. In the main Figure 6, panels B and C, we agree that for this reason the red labelled cell bodies likely include also subicular neurons, on the proximal side, in addition to L4 presubicular neurons. We now point out this caveat in the main text (line 324-326) and in the methods (line 591-592).

(1.2) Consistent with their subicular location, neuronal morphologies of the 'putative L4 cells' are selectively constrained within the subicular boundaries, i.e. they do not cross to the neighboring PrS (maybe a minor exception in Figs. 6_1D2,3). By definition, a neuron whose morphology is contained within a structure belongs to that structure.

From a functional point of view, for the HD system, the most important criterion for defining presubicular layer 4 neurons is their axonal projection to the LMN (Yoder and Taube 2011). From an electrophysiological standpoint, it is the capacity of layer 4 neurons to fire initial bursts (Simonnet et al., 2013; Huang et al., 2017). Anatomically, we note that the expectation that the apical dendrite should go straight up into layer 3 might not be a defining criterion in this curved and transitional periarchicortex. Presubicular layer 4 apical dendrites may cross through layer 3 and exit to the side, towards the subiculum (This is the red dendritic staining at the proximal end of the subiculum, at the frontier with the subiculum, Figure 6 C).

(1.3) As acknowledged by the authors in the discussion (line 408): the PrS is classically defined by the innervation domain of AD fibers. As Figure 6B clearly indicates, the retrogradely-labelled cells ('putative L4') are convincingly outside the input domain of the AD; hence, they do not belong to the PrS.

The reviewer is mistaken here, the deep layers 4 and 5/6 indeed do not lie in the zone innervated by the thalamic fibers (Simonnet et al., 2017; Nassar et al., 2018; Simonnet and Fricker, 2018) but still belong to the presubiculum. The presubicular deep layers are located below the superficial layers, next to, and in continuation of the subiculum. This is in agreement with work by Yoder and Taube 2011; Ishihara and Fukuda 2016; Boccara, … Witter, 2015; Peng et al., 2017 (Fig 2D); Yoshiko Honda et al., (Marmoset, Fig 2A) 2022; Balsamo et al., 2022 (Figure 2B).

(1.4) Along with the above comment: in my view, the optogenetic stimulation experiments are an additional confirmation that the 'putative L4 cells' are subicular neurons, since they do not receive AD inputs at all (hence, they are outside of the PrS); they are instead only indirectly driven upon strong excitation of the PrS. This indirect activation is likely to occur via PrS-to-Subiculum 'back-projections', the existence of which is documented in the literature and also nicely shown by the authors (see Figure 1_1 and line 109).

See above. Only superficial layers 1-3 of the presubiculum receive direct AD input.

(1.5) The electrophysiological properties of the 'putative L4 cells' are consistent with their subicular identity, i.e. they show a sag current and they are intrinsically bursty.

Presubicular layer 4 cells also show bursting behaviour and a sag current (Simonnet et al., 2013; Huang et al., 2017).

From the above considerations, and the data provided by the authors, I believe that the most parsimonious explanation is that these retrogradely-labelled neurons (from mammillary body injections), referred to by the authors as 'L4 PrS cells', are indeed pyramidal neurons from the distal subiculum.

We agree that the retrograde labeling is likely not limited to the presubicular layer 4 cells, and we now indicate this in the text (line 324-326). However, the portion of retrogradely labeled neurons that is directly below the layer 3 should be considered as part of the presubiculum.

I believe this is a fundamental issue that deserves clarification, in order to avoid confusion/misunderstandings in the field. Given the evidence provided, I believe that it would be inaccurate to call these cells 'L4 PrS neurons'. However, I acknowledge the fact that it might be difficult to convincingly and satisfactorily address this issue within the framework of a revision. For example, it is possible that these 'putative L4 cells' might be retrogradely-labelled from the Medial Mammillary Body (a major subicular target) since it is difficult to selectively restrict the injection to the LMN, unless a suitable driver line is used (if available). The authors should also consider the possibility of removing this subset of data (referring to putative L4), and instead focus on the rest of the story (referring to L3)- which I think by itself, still provides sufficient advance.

We agree with the reviewer that it is difficult to provide a satisfactory answer. To some extent, the reviewer’s comments target the nomenclature of the subicular region. This transitional region between the hippocampus and the entorhinal cortex has been notoriously ill defined, and the criteria are somewhat arbitrary for determining exactly where to draw the line. Based on the thalamic projection, presubicular layers 1-3 can now be precisely outlined, thanks to the use of viral labeling. But the presubicular layer 4 had been considered to be cell-free in early works, and termed ‘lamina dissecans’ (Boccara 2010), as the limit between the superficial and deep layers. Then it became of great interest to us and to the field, when the PrS layer 4 cells were first identified as LMN projecting neurons (Yoder and Taube 2011). This unique back-projection to the upstream region of the HD system is functionally very important, closing the loop of the Papez circuit (mammillary bodies - thalamus - hippocampal structures).

We note that the reviewer does not doubt our results, rather questions the naming conventions. We therefore maintain our data. We agree that in the future a genetically defined mouse line would help to better pin down this specific neuronal population.

We thank the reviewer for sharing their concerns and giving us the opportunity to clarify our experimental approach to target the presubicular layer 4. We hope that these explanations will be helpful to the readers of eLife as well.

(2) The PrS anatomy could be better clarified, especially in relation to its modular organization (see e.g. Preston-Ferrer et al., 2016; Ray et al., 2017; Balsamo et al., 2022). The authors present horizontal slices, where cortical modularity is difficult to visualize and assess (tangential sections are typically used for this purpose, as in classical work from e.g. barrel cortex). I am not asking the authors to validate their observations in tangential sections, but just to be aware that cortical modules might not be immediately (or clearly) apparent, depending on the section orientation and thickness. The authors state that AD fibers were 'not homogeneously distributed' in L3 (line 135) and refer to 'patches of higher density in deep L3' (line 136). These statements are difficult to support unless more convincing anatomy and . I see some L3 inhomogeneity in the green channel in Fig. 1G (last two panels) and also in Fig. 1K, but this seems to be rather upper L3. I wonder how consistent the pattern is across different injections and at what dorsoventral levels this L3 modularity is observed (I think sagittal sections might be helpful). If validated, these observations could point to the existence of non-homogeneous AD innervation domains in L3 - hinting at possible heterogeneity among the L3 pyramidal cell targets. Notably, modularity in L2 and L1 is not referred to. The authors state that AD inputs 'avoid L2' (line 131) but this statement is not in line with recent work (cited above) and is also not in line with their anatomy data in Fig. 1G, where modularity is already quite apparent in L2 (i.e. there are territories avoided by the AD fibers in L2) and in L1 (see for example the last image in Fig. 1G). This is the case also for the RSC axons (Fig. 1H) where a patchy pattern is quite clear in L1 (see the last image in panel H). Higher-mag pictures might be helpful here. These qualitative observations imply that AD and RSC axons probably bear a precise structural relationship relative to each other, and relative to the calbindin patch/matrix PrS organization that has been previously described. I am not asking the authors to address these aspects experimentally, since the main focus of their study is on L3, where RSC/AD inputs largely converge. Better anatomy pictures would be helpful, or at least a better integration of the authors' (qualitative) observations within the existing literature. Moreover, the authors' calbindin staining in Fig. 1K is not particularly informative. Subicular, PaS, MEC, and PrS borders should be annotated, and higher-resolution images could be provided. The authors should also check the staining: MEC appears to be blank but is known to strongly express calb1 in L2 (see 'island' by Kitamura et al., Ray et al., Science 2014; Ray et al., frontiers 2017). As additional validation for the staining: I would expect that the empty L2 patches in Figs. 1G (last two panels) would stain positive for Calbindin, as in previous work (Balsamo et al. 2022).

We now provide a new figure showing the pattern of AD innervation in PrS superficial layers 1 to 3, with different dorso-ventral levels and higher magnification (Figure 2). Because our work was aimed at identifying connectivity between long-range inputs and presubicular neurons, we chose to work with horizontal sections that preserve well the majority of the apical dendrites of presubicular pyramidal neurons. We feel it is enriching for the presubicular literature to show the cytoarchitecture from different angles and to show patchiness in horizontal sections. The non-homogeneous AD innervation domains (‘microdomains’) in L3 were consistently observed across different injections in different animals.

**Author response image 1. sa4fig1:** Thalamic fiber innervation pattern. (**A**), ventral, and (**B**), dorsal horizontal section of the Presubiculum containing ATN axons expressing GFP. Patches of high density of ATN axonal ramifications in L3 are indicated as “ATN microdomains”. Layers 1, 2, 3, 4, 5/6 are indicated. (**C**), High magnification image (63x optical section)(different animal).

We also provide a supplementary figure with images of horizontal sections of calbindin staining in PrS, with a larger crop, for the reviewer to check (Figure 3, see below). We thank the reviewer for pointing out recent studies using tangential sections. Our results agree with the previous observation that AD axons are found in calbindin negative territories (cf Fig 1K). Calbindin+ labeling is visible in the PrS layer 2 as well as in some patches in the MEC (Figure 3 panel A). Calbindin staining tends to not overlap with the territories of ATN axonal ramification. We indicate the inhomogeneities of anterior thalamic innervation that form “microdomains” of high density of green labeled fibers, located in layer 1 and layer 3 (Figure 3, Panel A, middle). Panel B shows another view of a more dorsal horizontal section of the PrS, with higher magnification, with a big Calbindin+ patch near the parasubiculum.

The “ATN+ microdomains” possess a high density of axonal ramifications from ATN, and have been previously documented in the literature. They are consistently present. Our group had shown them in the article by Nassar et al., 2018, at different dorsoventral levels (Fig 1 C (dorsal) and 1D (ventral) PrS). See also Simonnet et al., 2017, Fig 2B, for an illustration of the typical variations in densities of thalamic fibers, and supplementary Figure 1D. Also Jiayan Liu et al., 2021 (Figure 2 and Fig 5) show these characteristic microzones of dense thalamic axonal ramifications, with more or less intense signals across layers 1, 2, and 3. While it is correct that thalamic axons can be seen to cross layer 2 to ramify in layer 1, we maintain that AD axons typically do not ramify in layer 2. We modify the text to say, “mostly” avoiding L2 (line 130).

The reviewer is correct in pointing out that the 'patches of higher density in deep L3' are not only in the deep L3, as in the first panel in Fig 1G, but in the more dorsal sections they are also found in the upper L3. We change the text accordingly (line 135-136) and we provide the layer annotation in Figure 1G. We further agree with the reviewer that RSC axons also present a patchy innervation pattern. We add this observation in the text (line 144).

It is yet unclear whether anatomical microzones of dense ATN axon ramifications in L3 might fulfill the criteria of a functional modularity, as it is the case for the calbindin patch/matrix PrS organization (Balsamo et al., 2022). As the reviewer points out, this will require more information on the precise structural relationship of AD and RSC axons relative to each other, as well as functional studies. Interestingly, we note a degree of variation in the amplitudes of oEPSC from different L3 neurons (Fig. 2F, discussion line 420; 428), which might be a reflection of the local anatomo-functional micro-organization.

Minor points:(1) The pattern or retrograde labelling, or at least the way is referred to in the results (lines 104ff), seems to imply some topography of AD-to-PreS projections. Is it the case? How consistent are these patterns across experiments, and individual injections? Was there variability in injection sites along the dorso-ventral and possibly antero-posterior PrS axes, which could account for a possibly topographical AD-to-PrS input pattern? It would be nice to see a DAPI signal in Fig. 1B since the AD stands out quite clearly in DAPI (Nissl) alone.

Yes, we find a consistent topography for the AD-to-PrS projection, for similar injection sites in the presubiculum. The coordinates for retrograde labeling were as indicated -4.06 (AP), 2.00 (ML) and -2.15 mm (DV) such that we cannot report on possible variations for different injection sites.

(2) Fig. 2_2KM: this figure seems to show the only difference the authors found between AD and RS input properties. The authors could consider moving these data into main Fig. 2 (or exchanging them with some of the panels in F-O, which instead show no difference between AD and RSC). Asterisks/stats significance is not visible in M.

For space reasons we leave the panels of Fig. 2_2KM in the supplementary section. We increased the size of the asterisk in M.

(3) The data in Fig. 1_1 are quite interesting, since some of the PrS projection targets are 'non-canonical'. Maybe the authors could consider showing some injection sites, and some fluorescence images, in addition to the schematics. Maybe the authors could acknowledge that some of these projection targets are 'putative' unless independently verified by e.g. retrograde labeling. Unspecific white matter labelling and/or spillover is always a potential concern.

We now include the image of the injection site for data in Fig. 1_1 as a supplementary Fig. 1_2. The Figure 1_1 shows the retrogradely labeled upstream areas of Presubiculum.

**Author response image 2. sa4fig2:** Retrobeads were injected in the right Presubiculum.

(4) The authors speculate that the near-coincident summation of RS + AD inputs in L3 cells could be a potential mechanism for the binding of visual + HD information in PrS. However, landmarks are learned, and learning typically implies long-term plasticity. As the authors acknowledge in the discussion (lines 493ff) GluR1 is not expressed in PrS cells. What alternative mechanics could the authors envision? How could the landmark-update process occur in PrS, if is not locally stored? RSC could also be involved (Jakob et al) as acknowledged in the introduction - the authors should keep this possibility open also in the discussion.

A similar point has been raised by Reviewer 1, please check our answer to their point 2. Briefly, our results indicate that HD-to-landmark updating is a multi-step process. RSC may be one of the places where landmarks are learned. The subsequent temporal mapping of HD to landmark signals in PrS might be plasticity-free, as matching directional with visual landmark information based on temporal coincidence does not necessarily require synaptic plasticity. It seems likely that there is no local storage and no change in synaptic weights in PrS. The landmark-anchored HD signals reach LMN via L4 neurons, sculpting network dynamics across the Papez circuit. One possibility is that the trace of a landmark that matches HD may be stored as patterns of neural activity that could guide navigation (cf. El-Gaby et al., 2024, Nature) Clearly more work is needed to understand how the HD attractor is updated on a mechanistic level. Recent work in prefrontal cortex mentions “activity slots” and delineates algorithms for dynamic control of cognitive maps without synaptic plasticity (Whittington et al., 2025, Neuron): information may be stored in neural attractor activity, and the idea that working memory may rely on recurrent updates of neural activity might generalize to the HD system. We include these considerations in the discussion (line 499-503; 523-533) and also point to alternative models (line 518 -522) including modeling work in the retrosplenial cortex.

(5) The authors state that (lines 210ff) their cluster analysis 'provided no evidence for subpopulations of layer 3 cells (but see Balsamo et al., 2022)' implying an inconsistency; however, Balsamo et al also showed that the (in vivo) ephys properties of the two HD cell 'types' are virtually identical, which is in line with the 'homogeneity' of L3 ephys properties (in slice) in the authors' data. Regarding the possible heterogeneity of L3 cells: the authors report inhomogeneous AD innervation domains in L3 (see also main comment 2) and differences in input summation (some L3 cells integrate linearly, some supra-linearly; lines 272) which by itself might already imply some heterogeneity. I would therefore suggest rewording the statements to clarify what the lack of heterogeneity refers to.

We agree. In line 212 we now state “cluster analysis (Figure 2D) provided no evidence for subpopulations of layer 3 cells in terms of intrinsic electrophysiological properties (see also Balsamo et al., 2022).”

(6) n=6 co-recorded pairs are mentioned at line 348, but n=9 at line 366. Are these numbers referring to the same dataset? Please correct or clarify

Line 349 refers to a set of 6 co-recorded pairs (n=12 neurons) in double injected mice with Chronos injected in ATN and Chrimson in RSC (cf. Fig. 7E). The 9 pairs mentioned in line 367 refer to another type of experiment where we stimulated layer 3 neurons by depolarizing them to induce action potential firing while recording neighboring layer 4 neurons to assess connectivity. Line 367 now reads: “In n = 9 paired recordings, we did not detect functional synapses between layer 3 and layer 4 neurons.”

**Reviewer #3 (Recommendations For The Authors):**
Questions for the authors/points for addressing:I found that the slice electrophysiology experiments were not reported with sufficient detail. For example, in Figure 2, I am assuming that the voltage clamp experiments were carried out using the Cs-based recording solution, while the current clamp experiments were carried out using the K-Gluc intracellular solution. However, this is not explicitly stated and it is possible that all of these experiments were performed using the K-Gluc solution, which would give slightly odd EPSCs due to incomplete space/voltage clamp. Furthermore, the method states that gabazine was used to block GABA(A) receptor-mediated currents, but not when this occurred. Was GABAergic neurotransmission blocked for all measurements of EPSC magnitude/dynamics? If so, why not block GABA(B) receptors? If not blocking GABAergic transmission for measuring EPSCs, why not? This should be stated explicitly either way.

The addition of drugs or difference of solution is indicated in the figure legend and/or in the figure itself, as well as in the methods. We now state explicitly: “In a subset of experiments, the following drugs were used to modulate the responses to optogenetic stimulations; the presence of these drugs is indicated in the figure and figure legend, whenever applicable.” (line 632). A Cs-based internal solution and gabazine were used in Figure 5, this is now indicated in the Methods section (line 626). All other experiments were performed using K-Gluc as an internal solution and ACSF.

Methods: The experiments involving animals are incompletely reported. For example, were both sexes used? The methods state "Experiments were performed on wild‐type and transgenic C57Bl6 mice" - what transgenic mice were used and why is this not reported in detail (strain, etc)? I would refer the authors to the ARRIVE guidelines for reporting in vivo experiments in a reproducible manner (https://arriveguidelines.org/).

We now added this information in the methods section, subsection “Animals” (line 566-567). Animals of both sexes were used. The only transgenic mouse line used was the Ai14 reporter line (no phenotype), depending on the availability in our animal facility.

For experiments comparing ATN and RSC inputs onto the same neuron (e.g. Figure 2 supplement 2 G - J), are the authors certain that the observed differences (e.g. rise time and paired-pulse facilitation on the ATN input) are due to differences in the synapses and not a result of different responses of the opsins? Refer to https://pubmed.ncbi.nlm.nih.gov/31822522/ from Jess Cardin's lab. This could easily be tested by switching which opsin is injected into which nucleus (a fair amount of extra work) or comparing the Chrimson synaptic responses with those evoked using Chronos on the same projection, as used in Figure 2 (quite easy as authors should already have the data).

We actually did switch the opsins across the two injection sites. In Figure 2 - supplement 2G-J, the values linked by a dashed line result from recordings in the switched configuration with respect to the original configuration (in full lines, Chronos injected in RSC and Chrimson in ATN). The values from switched configuration followed the trend of the main configuration and were not statistically different (Mann-Whitney U test).

Statistical reporting: While the number of cells is generally reported for experiments, the number of slices and animals is not. While slice ephys often treat cells as individual biological replicates, this is not entirely appropriate as it could be argued that multiple cells from a single animal are not independent samples (some sort of mixed effects model that accounts for animals as a random effect would be better). For the experiments in the manuscript, I don't think this is necessary, but it would certainly reassure the reader to report how many animals/slices each dataset came from. At a bare minimum, one would want any dataset to be taken from at least 3 animals from 2 different litters, regardless of how many cells are in there.

Our slice electrophysiology experiments include data from 38 successfully injected animals: 14 animals injected in ATN, 20 animals injected in RSC, and 4 double injected animals. Typically, we recorded 1 to 3 cells per slice. We now include this information in the text or in the figure legends (line 159, 160, 297, 767, 826, 831, 832, 839, 845, 901, 941).

For the optogenetic experiments looking at the summation of EPSPs (e.g. figure 4), I have two questions: why were EPSPs measured and not EPSCs? The latter would be expected to give a better readout of AMPA receptor-mediated synaptic currents. And secondly, why was 20 Hz stimulation used for these experiments? One might expect theta stimulation to be a more physiologically-relevant frequency of stimulation for comparing ATN and RSC inputs to single neurons, given the relevance with spatial navigation and that the paper's conclusions were based around the head direction system. Similarly, gamma stimulation may also have been informative. Did the authors try different frequencies of stimulation?

Question 1. The current clamp configuration allows to measure EPSPamplification/prolongation by NMDA or persistent Na currents (cf. Fricker and Miles 2000), which might contribute to supralinearity.

Question 2. In a previous study from our group about the AD to PrS connection (Nassar et al., 2018), no significant difference was observed on the dynamics of EPSCs between stimulations at 10 Hz versus 30 Hz. Therefore we chose 20 Hz. This value is in the range of HD cell firing (Taube 1995, 1998 (peak firing rates, 18 to 24 spikes/sec in RSC; 41 spikes/sec in AD)(mean firing rates might be lower), Blair and Sharp 1995). In hindsight, we agree that it would have been useful to include 8Hz or 40Hz stimulations.

The GABA(A) antagonist experiments in Figure 5 are interesting but I have concerns about the statistical power of these experiments - n of 3 is absolutely borderline for being able to draw meaningful conclusions, especially if this small sample of cells came from just 1 or 2 animals. The number of animals used should be stated and/or caution should be applied when considering the potential mechanisms of supralinear summation of EPSPs. It looks like the slight delay in RSC input EPSP relative to ATN that was in earlier figures is not present here - could this be the loss of feedforward inhibition?

The current clamp experiments in the presence of QX314 and a Cs gluconate based internal solution were preceded by initial experiments using puff applications of glutamate to the recorded neurons (not shown). Results from those experiments had pointed towards a role for TTX resistant sodium currents and for NMDA receptor activation as a factor favoring the amplification and prolongation of glutamate induced events. They inspired the design of the dual wavelength stimulation experiments shown in Figure 5, and oriented our discussion of the results. We agree of course that more work is required to dissect the role of disinhibition for EPSP amplification. This is however beyond the present study.

Concerning the EPSP onset delays following RSC input stimulation: In this set of experiments, we compensated for the notoriously longer delay to EPSP onset, following RSC axon stimulation, by shifting the photostimulation (red) of RSC fibers to -2 ms, relative to the onset of photostimulation of ATN fibers (blue). This experimental trick led to an improved alignment of the onset of the postsynaptic response, as shown in the figure below for the reviewer.

**Author response image 3. sa4fig3:** In these experiments, the onset of RSC photostimulation was shifted forward in time by -2 ms, in an attempt to better align the EPSP onset to the one evoked by ATN stimulation.

We insert in the results a sentence to indicate that experiments illustrated in Figure 5 were performed in only a small sample of 3 cells that came from 2 mice (line 297), so caution should be applied. In the discussion we formulate more carefully, “From a small sample of cells it appears that EPSP amplification may be facilitated by a reduction in synaptic inhibition (n = 3; Figure 5)” (line 487).

Figure 7: I appreciate the difficulties in making dual recordings from older animals, but no conclusion about the RSC input can legitimately be made with n=1.

Agreed. We want to avoid any overinterpretation, and point out in the results section that the RSC stimulation data is from a single cell pair. The sentence now reads : “... layer 4 neurons occurred after firing in the layer 3 neuron, following ATN afferent stimuli, in 4 out of 5 cell pairs. We also observed this sequence when RSC input was activated, in one tested pair.” line (347-349)

Minor points:Line 104: 'within the two subnuclei that form the anterior thalamus' - the ATN actually has three subdivisions (AD, AV, AM) so this should state 'two of the three nuclei that form the anterior thalamus...'

Corrected, line 103

Line 125: should read "figure 1F" and not "figure 2F".

Corrected, line 124

Line 277-280: Why were two different posthoc tests used on the same data in Figures 3E & F?

We used Sidak’s multicomparison test to compare each event Sum vs. Dual (two different configurations at each time point - asterisks) and Friedman’s and Dunn’s to compare the nth EPSP amplitude to the first one for Dual events (same configuration between time points - hashmarks). We give two-way ANOVA results in the legend.